# CoNavBench: Collaborative Long-Horizon Vision-Language Navigation Benchmark

**Tianhang Wang**[1,2], **Xinhai Li**[3], **Fan Lu**[1], **Tianshi Gong**[3], **Jiankun Dong**[3],
**Weiyi Xue**[1], **Sanqing Qu**[1], **Chenjia Bai**[3], **Guang Chen**[1,2,4,✉,†]

[1]Tongji University [2]Shanghai Innovation Institution [3]Institute of Artificial Intelligence,
China Telecom [4] Shanghai Westwell Technology Co., Ltd

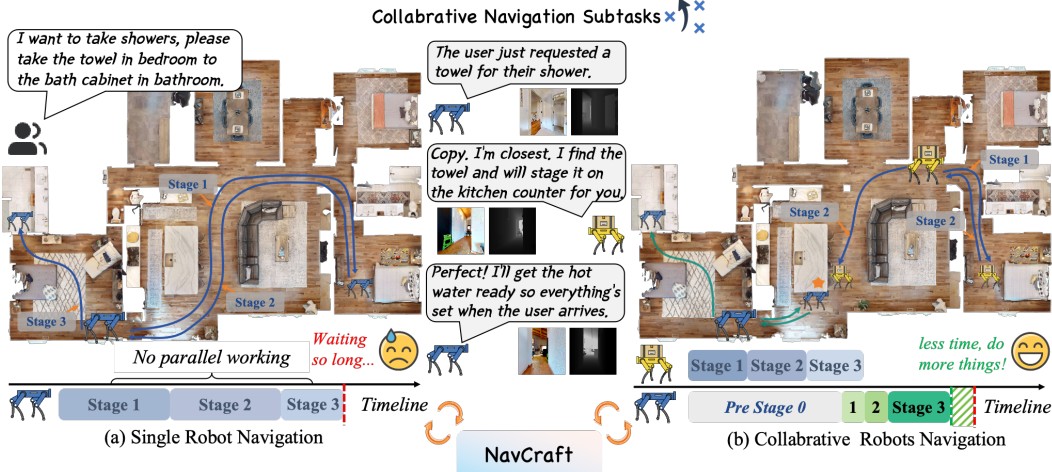

Figure 1: Comparison of single-robot and collaborative multi-robot navigation. (a) In the single-robot case, the agent must complete all stages sequentially, resulting in long delays and idle waiting. (b) In the collaborative case, subtasks are distributed across robots and executed in parallel, reducing overall completion time and enabling the team to accomplish more within the same horizon.

## ABSTRACT

Vision-and-Language Navigation (VLN) primarily focuses on a single-agent-centric approach that executes human instructions step-by-step. In real environments with high demand or parallel workflows, collaboration VLN offers distinct benefits including shorter makespan and greater robustness through parallelism and role specialization. Collaboration VLN also brings new challenges including congestion, handoff errors, and rendezvous timing, which single-agent formulations overlook. Current datasets and protocols remain single-agent centered, which hides opportunities for assistance and ignores inter-robot interference. We fill this gap with Collaborative Long-Horizon VLN benchmark (**CoNavBench**), consisting of 4048 single and collaborative episodes with graph-level annotations and a collaboration type taxonomy that controls handoff styles and rendezvous patterns. To generate and evaluate at scale, we build **NavCraft**, an automated graph-grounded data generation platform. A two-stage hierarchical agent first produces a long-horizon base mission for the primary robot and then instantiates helper robots, allocates subgoals, and specifies validated handoffs and rendezvous. The agents operate with a scene graph in the loop derived from Habitat-Sim, which enables reachability checks, travel time, and interference assessment, and iterative schedule repair via an efficiency tool library. As a reference, we provide a collaborative baseline based on a finetuned Qwen2.5-VL-3B. Trained with CoNavBench, collaborative policies reduce makespan and improve reliability over strong single robot counterparts, yielding **18.11%** step level success. CoNavBench.

---

✉ Corresponding author. Email: guangchen@tongji.edu.cn, † Project leader, supervised the work and defined the conceptualization.

Table 1: Comparison to VLN benchmarks. * The scale of our released benchmark is 4048, however NavCraft is able to generate unlimited data to be tested.

| Benchmark | Simulator | Task Type | Single Task | Collab Task | | Total Task |
|---|---|---|---|---|---|---|
| | | | | Num | Avg. Gain | |
| R2R | Matterport3D | Step-by-step Nav | 21567 | - | - | 21567 |
| REVERIE | Matterport3D | Obj Nav | 21702 | - | - | 21702 |
| VLN-CE | Habitat | Step-by-step Nav | 4475 | - | - | 4475 |
| FAO | Matterport3D | Obj Nav | 3848 | - | - | 3848 |
| Behavior-1k | OmniGibson | Complex Housework | 1000 | - | - | 1000 |
| IVLN | M3D&Habitat | Iterative VLN | 789 | - | - | 789 |
| Goat-Bench | Habitat | Iterative VLN | 725360 | - | - | 725360 |
| LHPR-VLN | Habitat | Multi-stage VLN | 3260 | - | - | 3260 |
| CoNavBench | Habitat | Multi-agent VLN | 2436* | 1612* | 21.08% | 4048* |

# 1 INTRODUCTION

Vision-and-Language Navigation (VLN) (Anderson et al., 2018; Thomason et al., 2019; Hong et al., 2020) has advanced from stepwise waypoint following to long-horizon, multi-stage settings that demand persistent reasoning and continual re-planning (Khanna et al., 2024). However, most formulations and datasets still assume a single robot agent, suppressing parallelism and ignoring inter-robot interference. In contrast, *collaborative* VLN views multiple robots as a coordinated team that exploits parallelism, anticipates and mitigates inter-robot interference, and optimizes wall-clock time and energy, which are central to user experience in real deployments (Puig et al., 2023).

To close this gap, we introduce Collaborative Long-Horizon VLN benchmark **CoNavBench**, to our knowledge the first systematic benchmark for collaborative VLN. CoNavBench comprises 4048 single- and multi-robot episodes together with a collaboration type taxonomy controlling handoff styles and rendezvous patterns. Each episode pairs long-horizon instructions with graph-level annotations, enabling efficient metrics including task success, makespan, and interference time. Given a long-range instruction, a team must decompose the mission, assign roles, and coordinate handoffs or rendezvous to minimize time while avoiding congestion.

We identify three escalating challenges for collaborative VLN: (i) **Cooperation-ready task synthesis**: constructing long-horizon single-robot base tasks with explicit stage boundaries and cross-room dependencies that create genuine opportunities for assistance (Xu et al., 2022); (ii) **Conflict-free team scheduling**: lifting a base task into a multi-robot schedule with role assignments, temporal ordering, and rendezvous; and (iii) **In-loop efficiency optimization**: given a feasible schedule, estimating team-level time, anticipating bottlenecks, and issuing actionable guidance.

To address the above challenges, we present **NavCraft**, a graph-grounded generation platform for CoNavBench. NavCraft first constructs a semantically augmented scene graph from Habitat-Sim (Savva et al., 2019) as the planning blueprint (Rana et al., 2023). Each node is labeled via hierarchical clustering to assign room categories and functional properties, and edges encode topology and traversability. Over this spatially grounded representation, a two-stage hierarchical agent operates: *NavCraft-S* produces a long-horizon single-robot base plan with cross-room scope and explicit stage structure, and *NavCraft-C* lifts it into a collaborative schedule by instantiating helper robots, allocating subgoals, and validating handoffs and rendezvous. Unlike text-only prompting (White et al., 2023), which lacks spatial grounding, and asset-specific simulators (Yang et al., 2024), which limit throughput and versatility. NavCraft enables context-aware task generation conditioned on user profiles (Wang et al., 2025b) and robot capabilities, improving diversity and schedule validity.

We further propose an **on-graph efficiency tool library** that unifies validation and guidance within the scene-graph loop. The library translates language intents into numerical constraints over distances, widths, occlusions, and occupancy; verifies reachability, interference, and estimates time. It then issues recommendations for subgoal allocation, rendezvous timing, helper deployment, and route revision. The agent consumes these recommendations in a closed loop, preserving accuracy while avoiding full-physics rollouts at each step (Wang et al., 2025a).

Finally, we provide a reference stack coupling Qwen-series LLMs (Bai et al., 2025) with a memory-aware mechanism (Song et al., 2025). Policies trained on CoNavBench achieve **18.11%** step level success than single-robot, indicating a practical path toward deployable collaborative navigation.

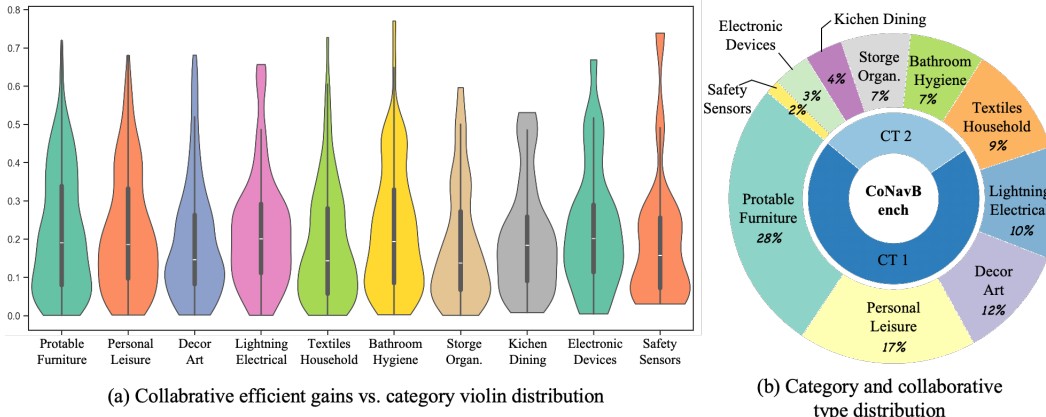

(a) Collabrative efficient gains vs. category violin distribution

(b) Category and collaborative type distribution

Figure 2: CoNavBench benchmark. **(a) Collaborative efficiency by category.** Violin plots show the distribution of category-wise *efficiency gain* over a single-robot baseline, yielding an average gain of 20% across categories. **(b) Category and collaboration-type distribution.** The benchmark covers a broad and balanced set of household target-object categories (outer ring) and two collaboration types (inner ring), evidencing rich object diversity that supports generalizable evaluation.

## 2 RELATED WORK

**Vision-and-Language Navigation** Embodied Vision-and-Language Navigation (VLN) studies language-conditioned navigation in complex environments. Methods are typically studied in discrete and continuous settings. In discrete VLN (Chen et al., 2022; Zhou et al., 2023), agents move on a fixed panoramic graph of predefined nodes. The abstraction emphasizes high-level decisions while masking metric geometry and collisions. Recent work introduces LLM-augmented planners with retrieval-augmented memory to improve instruction parsing and subgoal proposal (Chen et al., 2024; Zheng et al., 2024), yet these systems still assume oracle connectivity and lack low-level feasibility checks. In continuous environments (Dong et al., 2025), many approaches pretrain waypoint predictors to propose candidate positions for high-level planning (Qiao et al., 2025b; Shi et al., 2025), but such models often overfit and generalize poorly. End-to-end dual-system alternatives reduce this reliance: a high-level planner performs embodied planning with a slow/fast context (Wei et al., 2025), and a low-level controller utilizes a diffusion-policy (Cai et al., 2025) for local motion, improving responsiveness without scene-specific priors. However, across both settings, systems and benchmarks remain predominantly single-agent, with limited modeling of collaboration.

**Benchmark for Vision-and-Language Navigation** Progress in VLN has been driven by datasets (Qiao et al., 2025a) that steadily raise task complexity and evaluation fidelity. Early datasets, such as R2R (Anderson et al., 2018) and R4R (Jain et al., 2019), study step-by-step instruction following along predefined panoramic trajectories. VLN-CE (Krantz et al., 2020) shifts to continuous control in photorealistic simulators, emphasizing perception and low-level decision making. More recent datasets, including CVDN (Thomason et al., 2019), REVERIE (Qi et al., 2019), and SOON (Zhu et al., 2021), introduce dialogue history, object-centric grounding, and complex instruction comprehension. OVMM (Yenamandra et al., 2023) and Behavior-1K (Li et al., 2022) couple navigation with manipulation and interaction to approximate extended real-world workflows. IVLN (Krantz et al., 2022) and GOAT-Bench (Khanna et al., 2024) enable sequential multi-episode navigation with memory across independent goals, and LHPR-VLN (Song et al., 2025) targets long-horizon planning with multi-stage subtasks in complex indoor environments. Despite this progress, existing benchmarks remain single-agent and lack collaboration primitives (Wang et al., 2023a), which are necessary to study Collaborative VLN with multiple agents subtasks in highly complex environments. This gap motivates our collaborative long-horizon benchmark and platform.

**LLM Agents for Vision-and-Language Navigation** LLM agents are widely used as policies in interactive domains such as the Web (Chae et al., 2024), games (Hu et al., 2024a), robotics (Zitkovich et al., 2023; Wang et al., 2024b; Cheng et al., 2025; Jing et al., 2025), and design (Hu et al., 2024b), where they parse instructions, perceive scenes, and invoke tools. In VLN, these agents typically serve as navigators, grounding observations to produce subgoals or actions. Representative examples include VELMA (Schumann et al., 2023) in Street-View and indoor planners such as Nav-

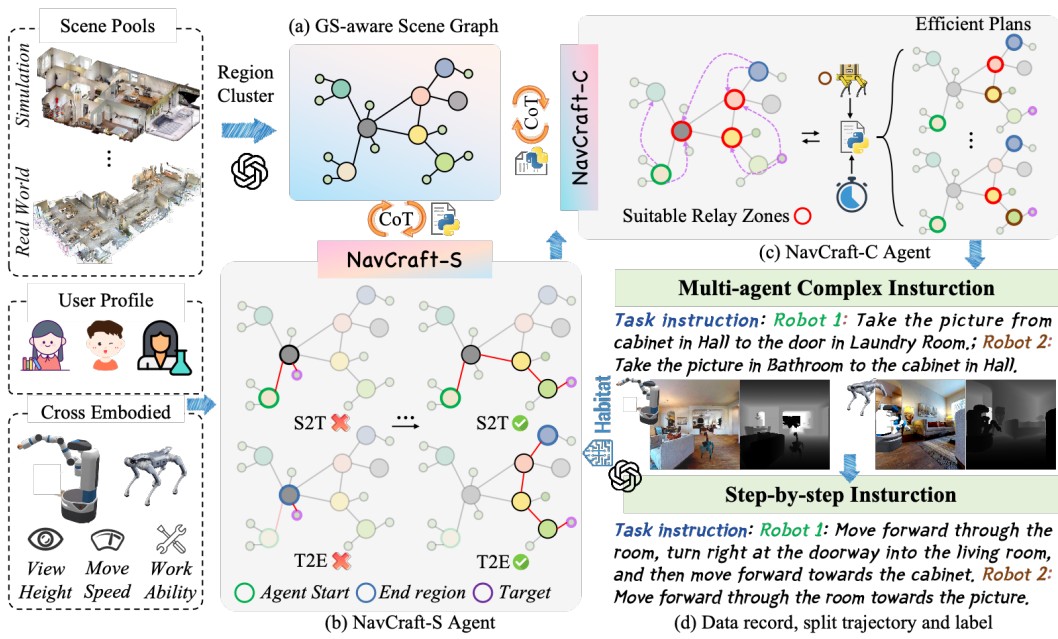

Figure 3: **NavCraft pipeline for CoNavBench benchmark data generation and scheduling.**

iLLM (Zheng et al., 2023) via prompts (Saravia, 2022). These systems are single-robot and are evaluated zero-shot or with fine-tuning on existing datasets. However, prior work treats the agent as a navigator (Wang et al., 2024a) rather than a generator. Our departure is to utilize the agent as a data generation and scheduling engine for CoNavBench: NavCraft's hierarchical agent synthesizes cooperation-ready long-horizon tasks, allocates roles, and produces team-aware schedules.

## 3 PLATFORM, BENCHMARK AND METRICS

We introduce NavCraft, a data-generation platform tailored for Collaborative VLN. Using NavCraft, we construct the CoNavBench benchmark, which enables systematic evaluation of models on long-horizon, multi-agent planning and execution within vision-language navigation.

### 3.1 NAVCRAFT

#### 3.1.1 SCENE GRAPH GENERATION

We annotate each node of Habitat connectivity graph $G = (V, E)$ (Wang et al., 2023b) with region types to obtain a semantic-aware graph. Each node $i \in V$ stores a 3D position $\mathbf{p}_i = [x_i, y_i, z_i]^\top$. And we also know each region object $m$'s 3D position $\mathbf{x}_m = [x_m, y_m, z_m]^\top$ and region type $r(m)$.

**Instance Proximal Voting** We seed each node by *object-centric* k-NN plurality to preserve coarse-grained cues. For node $i$, we search the $k$ nearest annotated objects on the ground plane and assign the plurality region:

$$\mathcal{N}_k(i) = \underset{m \in \mathcal{M}}{\arg \operatorname{topk}} \|\mathbf{p}_i - \mathbf{x}_m\|_2, \qquad \hat{r}_i^{(0)} = \arg \max_c \sum_{m \in \mathcal{N}_k(i)} \mathbf{1}[r(m) = c].$$

*where $\mathcal{M}$ is the set of all neighborhood objects, and $k=3$ by default.*

**Neighborhood Consensus** After seeding node labels with IPV, we observe occasional isolated mis-labels near narrow passages (e.g., doorways), where object-centric votes can flip a single node at region boundaries. To address this, we apply a targeted, local correction. A node is eligible for relabeling only if two guards hold simultaneously: (i) its graph degree is modest, $2 \le \deg(i) \le 4$, and (ii) its provisional label $\hat{r}_i^{(0)}$ disagrees with every neighbor $C(i)$ in $G$. If both conditions are

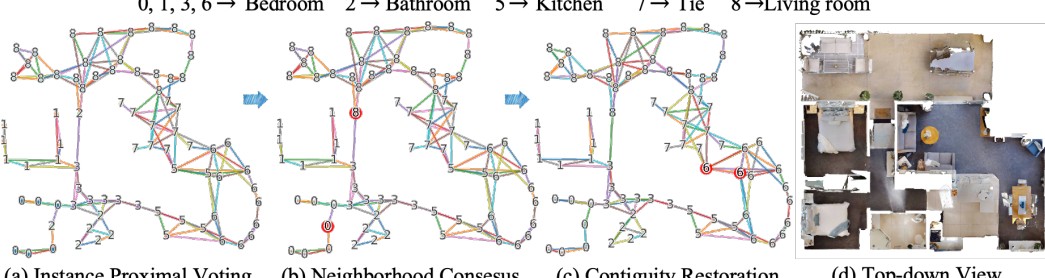

0, 1, 3, 6 → Bedroom    2 → Bathroom    5 → Kitchen    7 → Tie    8 → Living room

(a) Instance Proximal Voting    (b) Neighborhood Consesus    (c) Contiguity Restoration    (d) Top-down View

Figure 4: Visualization of the scene graph generation pipeline.

met, the node adopts the label of the nearest labeled neighbor on the navigation plane:

$$j^{\star} = \arg \min_{j \in C(i)} \|\mathbf{p}_i - \mathbf{p}_j\|_2, \qquad \hat{r}_i^{(1)} = \begin{cases} \hat{r}_{j^{\star}}^{(0)}, & \text{if } 2 \leq \deg(i) \leq 4 \text{ and } \forall j \in C(i) : \hat{r}_j^{(0)} \neq \hat{r}_i^{(0)}, \\ \hat{r}_i^{(0)}, & \text{otherwise.} \end{cases}$$

**Contiguity Restoration** After IPV and NC seeding, we still observe mislabeled *regions* caused by structural barriers such as walls, which fragment a class into multiple small, fractured islands. To restore semantic contiguity without excessive smoothing, we operate at the connected-component level. For each class $c$, form the induced subgraph $G_c = G[\{i : \hat{r}_i^{(1)} = c\}]$ and keep its largest component and process any remaining component $\mathcal{C}$ as follows. If $\mathcal{C}$ touches no other class, delete it; otherwise, reassign the entire component to the adjacent class with boundary nodes closest to the component centroid:

$$\boldsymbol{\mu}_{\mathcal{C}} = \frac{1}{|\mathcal{C}|} \sum_{i \in \mathcal{C}} \mathbf{p}_i, \qquad c^{\star} = \arg \min_{c'} \frac{1}{|B_{c'}|} \sum_{j \in B_{c'}} \|\mathbf{p}_j - \boldsymbol{\mu}_{\mathcal{C}}\|_2, \quad \hat{r}_i^{(2)} = c^{\star} \ \forall i \in \mathcal{C}.$$

*where* $B_{c'}$ *are boundary neighbors of* $\mathcal{C}$ *with class* $c'$ *in* $G$. As shown in Figure 4, the mislabeled '5-Kitchen' island is relabeled into '6-Bedroom' via contiguity restoration.

**Graph Contextual Typing** After the preceding steps, a small fraction of regions may still be typed as Unknown. Intuitively, these are ambiguous areas where local object votes and connectivity cues are not confident enough on their own. For any region type still `Unknown`, we make a single pass that combines graph context and object inventory. We build a compact summary: (i) the histogram of adjacent room types in $G$, and (ii) the top-5 object names in that region and query a lightweight instruction-following model $h_\phi$ for the most plausible type; otherwise, we keep the current label:

$$\hat{r}_i^{\text{final}} = \begin{cases} h_\phi(\text{adjacent-type hist, top-5 objects}), & \text{if } \hat{r}_i^{(2)} = \texttt{Unknown}, \\ \hat{r}_i^{(2)}, & \text{otherwise.} \end{cases}$$

*where* $h_\phi$ *is the* `GPT4o_mini` *and its output updates the per-scene region dictionary, while node positions and edges remain unchanged.*

### 3.1.2 NAVCRAFT-S

**Goal** Given the room-labeled connectivity graph and per-scene item lists, NavCraft-S selects a feasible triple: start region $s$, target-object region $t$, end region $e$ and validates it with graph constraints, and synthesizes a directed region-to-region path. A *user profile* $\pi$ is injected via the prompt as a *light preference* when sampling target objects, and it never overrides feasibility.

**Profile-conditioned sampling** We simulate user demand with a lightweight role profile $\pi$ that encodes age, occupation, and lifestyle. The profile is injected into the prompt to encourage diverse habits and phrasing, and it is used only as a tie-breaker when multiple objects or destinations are equally eligible. Following the role templates in NavRAG (Wang et al., 2025b), NAVCRAFT-S then samples a portable object and compatible start and end regions conditioned on $\pi$. This simple conditioning increases variety without relying on hand-crafted priors.

**Feasibility** Given a candidate triple (s,t,e), NavCraft-S must first ensure that the underlying navigation problem is meaningful: each leg should be reachable on the region graph and long enough to

span multiple rooms. As shown in Figure 3, we evaluate the two legs $s \to t$ and $t \to e$ on the region graph using the skill library. Let $L(u, v) = \text{dist}_H(u, v)$ denote hop distance on the region graph, $\text{conn}(u, v)$ indicate reachability, and $\text{adj}(u, v) \iff L(u, v) = 1$. We introduce a hop threshold $\tau \geq 1$ to control the minimum cross-room extent (CoNavBench sets $\tau = 2$). A leg is admissible if it is connected and non-trivial in length. To avoid redundancy with the non-adjacency rule, we enforce

$$\text{leg\_ok}(u, v) := \text{conn}(u, v) \wedge L(u, v) \geq \max\{2, \tau\},$$
$$\text{valid} := \text{leg\_ok}(s, t) \wedge \text{leg\_ok}(t, e).$$

In words, each leg must be reachable and span at least $\max\{2, \tau\}$ hops, and larger $\tau$ encourages longer-range plans. Once $\text{valid}$ holds, we concatenate hop-shortest paths for the two legs and compress consecutive nodes that belong to the same region to obtain region transitions:

$$\text{Path}(s \rightsquigarrow e) = \text{SP}_H(s, t) \oplus \text{SP}_H(t, e).$$

### 3.1.3 NAVCRAFT-C

Given the high-level triple $(s, t, e)$ and the region path produced by NAVCRAFT-S, NAVCRAFT-C lifts the single agent plan to a collaborative execution on the same semantic and geometric graph. The module first decides whether collaboration is beneficial. If collaboration is selected, it fixes a collaboration type and hands off. Details of skill templates appear in the Appendix.

**Type abstraction** We use two canonical handoff patterns that allow the planner to reason about cooperation independently of motion primitives. *Type A1*: the collaborator picks up the object in region $t$, hands off at transfer region $x$, and the main agent delivers from $x$ to $e$. *Type A2*: the main agent picks up in $t$, hands off at $x$, and the collaborator delivers from $x$ to $e$.

**Augmented metric graph** Let $\mathcal{T}$ be the set of region nodes and $\mathcal{A} = \{a_e, a_x\}$ the set of anchors (end asset, candidate transfer asset). For any anchor $a \in \mathcal{A}$, let $c(a) \in \mathcal{T}$ denote the region that contains $a$. We build an augmented node graph $G^+$ from the Habitat connectivity graph $G$. Edges in $G^+$ use 2D Euclidean weights. Each anchor $a$ is inserted at its physical location and linked to the nearest navigable node. We then use a single distance on $G^+$:

$$d(x, y) = \text{dist}_{G^+}(x, y).$$

**Collaborative Generation** We quantify when it is worthwhile to involve a second agent by comparing how much travel load the main agent would bear alone versus under different collaboration patterns. Intuitively, a collaboration is only accepted if bringing in the helper strictly shortens the main agent's own route. The single–agent baseline load borne by the main agent is:

$$C_{\text{solo}} = d(s, o) + d(o, a_e).$$

Given a candidate tuple $(type, a_x)$, we evaluate the main agent load under two types of collaboration:

$$J_{r_1}^{\text{A1}} = d(s, a_x) + d(a_x, a_e), \qquad J_{r_1}^{\text{A2}} = d(s, t) + d(t, a_x).$$

Here $J_{r_1}$ denotes the load borne by the main agent, and the A1/A2 indexes the collaboration types. We accept collaboration if:

$$\min\{J_{r_1}^{\text{A1}}, J_{r_1}^{\text{A2}}\} < C_{\text{solo}},$$

and we report the improvement ratio $\alpha = \min\{J_{r_1}^{\text{A1}}, J_{r_1}^{\text{A2}}\}/C_{\text{solo}}$. A candidate must satisfy the scene-graph guards: (i) $x \neq t$ and $a_x$ exists under $x$; (ii) both agents can reach $x$ in $G^+$; (iii) the collaborator's start follows non-adjacency and connectedness rules to $(t, x)$. The planner iteratively proposes tuples and records them with $\alpha < 1$.

### 3.2 THE CONAVBENCH BENCHMARK AND METRICS

**Benchmark Definition** The proposed CoNavBench benchmark is designed to evaluate collaborative embodied navigation under multi-agent settings. Unlike conventional single-agent VLN tasks, where an agent must complete a long-horizon instruction end-to-end, CoNavBench decomposes a complex instruction into multiple collaborative subtasks (referred to as *collab-stages*). A typical high level user command follows the pattern: *"Find object A at location X, and deliver it to location Y"*. Instead of requiring a single agent to traverse and manipulate across the entire trajectory, the

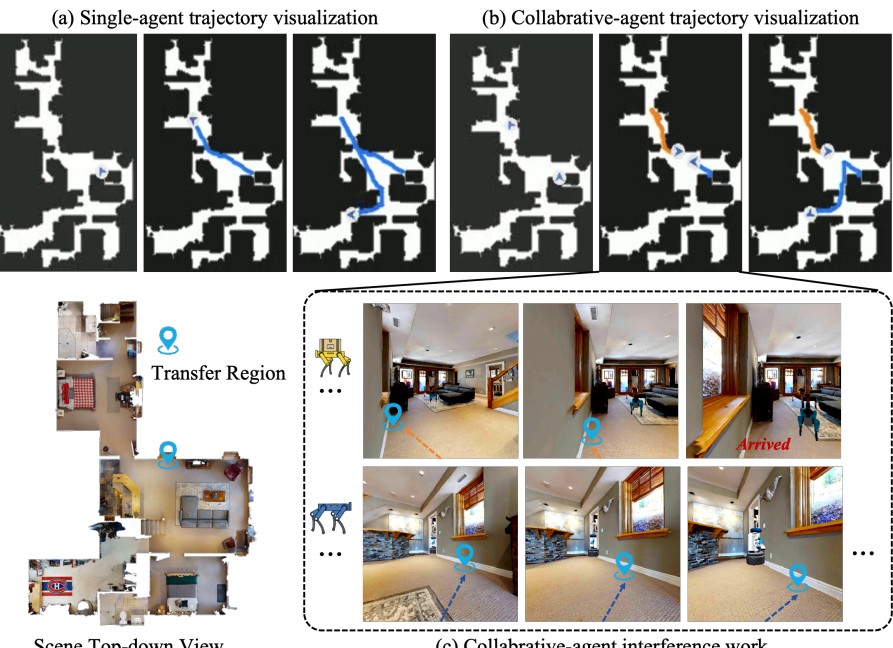

Figure 5: (a) Single-agent trajectory visualization: the agent navigates independently following the given instruction. (b) Collaborative-agent trajectory visualization: two agents cooperate during navigation, showing more efficient and coordinated exploration paths. (c) Collaborative-agent interference work: visual examples from the agents' first-person perspectives, illustrating interaction and coordination in shared environments.

task is distributed across two agents. Concretely, one agent is responsible for locating and transporting the target to an intermediate relay point, after which another agent continues the delivery to the final goal. As shown in Figure 5, this decomposition produces coordinated trajectories with a clear relay handoff between agents. Compared to the single-agent rollout in Figure 5.(a), the collaborative setting in Figure 5.(b) reduces backtracking and shortens paths by assigning complementary exploration regions. First-person views in Figure 5.(c) further illustrate the handoff and mutual avoidance behaviors that enable efficient, reliable delivery. This decomposition mirrors realistic multi-robot cooperation, mitigates memory overflow issues in long-horizon reasoning, and empirically improves task success rates. Meanwhile, we also follow (Song et al., 2025) to decompose the high-level tasks and create step-by-step VLN tasks for each trajectory segment to alleviate the inherent difficulty of executing the abstract instructions.

**Evaluation Metrics** To rigorously measure performance on CoNavBench, we employ standard navigation metrics: (i) **Success Rate (SR)**, the percentage of episodes where the agent(s) successfully complete the task within a 1.0 m goal threshold; (ii) **Success weighted by Path Length (SPL)**, which normalizes success by the efficiency of the trajectory; and (iii) **Navigation Error**, computed as the geodesic distance between the agent's final position and the goal when the task terminates. In addition, we extend evaluation with two subtask metrics originally proposed in LH-VLN (Song et al., 2025): **Independent Completion Rate (ICR)** quantifies the success of each sub-task individually, thereby providing insight into the robustness of agents when executing isolated segments of the collaborative pipeline. **Conditional Success Rate (CSR)** measures the overall success of the full multi-agent task, where completion depends on all preceding subtasks being successfully executed. CSR thus captures interdependencies across collab-stages and reflects the degree to which agents can coordinate seamlessly over extended task horizons.

## 4 EXPERIMENT

### 4.1 EXPERIMENTAL SETTING

**Simulator Settings** All experiments are conducted in HABITAT3 (Puig et al., 2023), a continuous 3D simulation platform for vision-and-language navigation (VLN). Unless otherwise specified, each

Table 2: Performance comparison on the **Single-Agent Task** in CoNavBench. Results are shown for both high-level tasks and step-by-step subtasks.

| Method | Type | Single Agent Task (High-level/Step-by-step Tasks) | | | | |
|---|---|---|---|---|---|---|
| | | SR↑ | SPL↑ | ISR↑ | CSR↑ | NE↓ |
| Random | - | 0.00/1.26 | 0.00/1.26 | 1.61/1.26 | 1.21/1.26 | 7.25/7.56 |
| Qwen2.5-VL-3B* | Zero-shot | 4.30/10.41 | 0.98/2.55 | 14.25/10.41 | 16.10/10.41 | 6.91/7.80 |
| | Finetuned | 12.90/29.65 | 6.08/13.81 | 23.92/29.65 | 26.22/29.65 | 6.40/6.74 |
| Qwen2.5-VL-7B* | Zero-shot | 0.00/1.26 | 0.00/1.26 | 1.84/1.26 | 1.33/1.26 | 7.21/7.56 |
| | Finetuned | 10.22/22.40 | 4.93/12.57 | 22.58/22.40 | 22.45/22.40 | 6.39/7.46 |

Table 3: Performance comparison on the **Collaborative-Agent Task** in CoNavBench. Results are shown for both high-level tasks and step-by-step subtasks.

| Method | Type | Collaborative Agent Task (High-level/Step-by-step Tasks) | | | | |
|---|---|---|---|---|---|---|
| | | SR↑ | SPL↑ | ISR↑ | CSR↑ | NE↓ |
| Random | - | 0.00/3.43 | 0.00/3.43 | 2.30/3.43 | 1.76/3.43 | 7.15/6.35 |
| Qwen2.5-VL-3B* | Zero-shot | 8.67/19.86 | 2.89/5.44 | 16.12/19.86 | 16.12/19.86 | 6.82/6.50 |
| | Finetuned | 11.11/35.02 | 4.82/16.88 | 20.19/35.02 | 20.12/35.02 | 6.55/5.79 |
| Qwen2.5-VL-7B* | Zero-shot | 0.00/3.61 | 0.00/3.61 | 1.90/3.61 | 1.36/3.61 | 7.12/6.35 |
| | Finetuned | 11.65/29.78 | 6.24/16.56 | 21.00/29.78 | 20.93/29.78 | 6.74/6.20 |

agent is equipped with synchronized RGB and depth sensors mounted in three directions: front, left $(+60°)$, and right $(-60°)$. To ensure comparability with prior work, we adopt the atomic action space used in LH-VLN (Song et al., 2025): *move forward* $(+0.25 \, \text{m})$, *turn left* $(+30°)$, *turn right* $(-30°)$, and *stop*. An episode (or sub-episode) is regarded as completed either when the agent issues *stop* or when the geodesic distance to the designated target falls below $1.0 \, \text{m}$.

**Embodied** We instantiate two articulated robot embodiments from URDF models: the Fetch mobile manipulator and Boston Dynamics Spot. Fetch features a wheeled base with an upper-body manipulator and structural frame; Spot is a quadruped robot capable of carrying a back-mounted arm. These embodiments allow us to test navigation and embodied interaction under heterogeneous morphology and locomotion dynamics, while keeping sensing and policy stacks consistent.

**Scene Assets** Our environments are primarily drawn from HM3D (Ramakrishnan et al., 2021), comprising 216 large-scale, semantically annotated indoor reconstructions. We adopt the scene graph initialization from SCALEVLN to provide object- and room-level structure. In addition, we develop a *real2sim* pipeline that scans real-world indoor spaces and imports them into Habitat, enabling closed-loop validation of our data generation with NAVCRAFT. More details are in the Appendix.

**Baselines and Training Settings** We follow a trajectory-supervised learning paradigm. For multimodal reasoning, we employ the Qwen-2.5VL family and report results for both small- and mid-scale variants (3B/7B), evaluated in zero-shot settings and after fine-tuning on the CoNavBench corpus. Visual features are extracted by a ViT backbone from EVA-CLIP-02-LARGE; the visual encoder remains *frozen* during all training runs to stabilize optimization and reduce compute. Unless noted otherwise, we fine-tune the language-conditioned policy and control heads (full-parameter fine-tuning on the non-visual modules), using Adam with a learning rate of $3 \times 10^{-5}$. Training is performed on four NVIDIA A800 GPUs with a per-step batch size of 1; a complete run typically finishes in about four days. We apply standard practices for reproducibility: fixed random seeds, gradient clipping, and validation-based early stopping. More details are in the Appendix.

## 4.2 RESULT AND ANALYSIS

**Single Agent Performance** Table 2 reports single-agent results under both the high-level tasks and step-by-step protocols. Random policies fail across all metrics, confirming the benchmark's non-trivial difficulty. Zero-shot Qwen2.5-VL models perform slightly above random on SR and ISR yet remain far from practical utility, with task-level SR below 5%, indicating that high-level instructions are hard to follow. After finetuning, both Qwen2.5-VL-3B and 7B show clear absolute gains:

Table 4: Performance, efficiency, cost and latency of NAVCRAFT-powered agents. Higher numbers are better (↑) except Cost (↓). Note that the success rate represents task generation.

| Powered Agent | Success Rate ↑ | | Collab Gain ↑ | | Cost ($) ↓ | Sample Eff. (s/iter)↓ |
|---|---|---|---|---|---|---|
| | Single | Collab | Max | Avg | | |
| *Google* | | | | | | |
| 2.0-flash | 47% | 8.51% | 29.72% | 22.32% | 0.265 | 16.82 |
| 2.5-flash-nothink | 41% | 9.76% | 13.18% | 9.24% | 0.902 | 18.71 |
| *Claude* | | | | | | |
| 3-5-haiku | 51% | 3.90% | 50.07% | 37.06% | 1.142 | 14.94 |
| *OpenAI* | | | | | | |
| 4o | 77% | 46.75% | 75.23% | 21.07% | 5.242 | 30.85 |
| 4o-mini | 64% | 26.56% | 47.32% | 25.12% | 0.360 | 21.41 |
| 4.1-mini | 68% | 23.53% | 42.80% | 19.10% | 0.494 | 17.92 |

SR improves at both the high-level and step-by-step settings, accompanied by consistent increases in SPL, ISR, and CSR, and a reduction in navigation error. Despite these improvements, absolute performance remains modest, underscoring that CoNavBench poses a challenging single-agent benchmark and leaving ample headroom for future methods.

**Collaborative Agent Performance** Table 3 compares collaboration with single-agent execution. Under the **step-by-step** protocol, both Qwen2.5-VL-3B and 7B improve: the finetuned 3B model raises SR from **29.65%** to **35.02%**, and the finetuned 7B model rises from **22.40%** to **29.78%** with SPL increasing from **12.57** to **16.56** and NE decreasing, consistent with CoNavBench's design that splits a long multi-stage instruction into single-stage subtasks and shortens the decision horizon. In the **high-level** end-to-end evaluation, gains are smaller and some metrics slightly regress, partly because test-time relay planning introduces intermediate handoff points and auxiliary phrasing that can cause mild vision–language mismatches, and partly because high-level completion is inherently difficult: success compounds across stages (CSR), early errors propagate, and coordination overhead from state synchronization and re-localization enlarges the effective search space. Finally, although the 7B model is competitive, it does not surpass 3B under the current data budget, which suggests under-training rather than a fundamental limitation; viewed in isolation, the collaborative setting still benefits 7B (SR **22.40%** to **29.78%**, SPL **12.57** to **16.56**, NE decreases), indicating that multi-agent decomposition reliably improves local competence even when model capacity is not fully exploited.

### 4.3 ABLATION STUDIES

Table 4 evaluates representative off-the-shelf agents from Google, Claude, and OpenAI under both single and collaborative settings, revealing clear capability and cost–efficiency trade-offs. Collaboration generally improves performance but with varying magnitude across families. Claude-3.5-haiku shows the second relative collaboration gain 50.07% despite very low absolute collaborative task generation success rate 3.90%, suggesting that weaker agents can benefit from task decomposition but still fail to achieve reliable success. OpenAI's 4o achieves the strongest absolute results with 77% in the single-agent and 46.75% success rate in the collaborative task generation, while Google's models provide lower cost per sample but limited task generation success rate, which indicates that efficiency alone cannot compensate for weak grounding ability. Among these choices, GPT-4o-mini offers the most favorable balance: it reaches 26.56% collaborative task generation SR and the highest average collaboration gains among the OpenAI variants 25.12% at substantially lower cost, about 0.360 per sample compared to 5.242 for 4o. Based on this analysis, we adopt GPT-4o-mini as the data generation agent for CoNavBench, allowing us to scale instruction and trajectory synthesis while maintaining a strong balance between quality and price efficiency.

**Limitation** Our framework still has several limitations. First, the data generation pipeline relies on GPT API models, which can introduce stylistic bias and make reproducibility sensitive to backend updates; a natural next step is to train or distill an open LLM specialized for CoNavBench and to improve throughput with batching, caching, and on-graph pruning. Second, NavCraft-C currently

targets two-agent relay patterns in indoor HM3D-style scenes, so coverage of richer collaboration and three or more robots remains limited, partly due to scene size.

## 5 CONCLUSION

We introduced **CoNavBench**, a collaborative vision-and-language navigation benchmark with 4048 episodes, a collaboration type taxonomy, and graph-level annotations that enable team-aware evaluation (success, makespan, energy, and interference time). To populate and study this setting, we presented **NavCraft**, a graph-grounded generation platform built on a semantically augmented scenegraph substrate, along with a two-stage agent (*NavCraft-S* and *NavCraft-C*) and an on-graph efficiency tool library for closed-loop validation and guidance. Beyond establishing the benchmark, we provide a reference stack that couples Qwen-series LLMs. Policies trained on CoNavBench achieve **18.11%** step-level task success rate compared to single-robot, indicating a practical path toward deployable collaborative vision-and-language navigation.

**Acknowledgments**: This work was supported by the National Key Research and Development Program of China (No. 2024YFE0211000), in part by the National Natural Science Foundation of China (No. 62372329, 62506263, 62506264), in part by the Shanghai Scientific Innovation Foundation (No. 23DZ1203400), in part by the China Postdoctoral Science Foundation (No. BX20250383, GZB20250385, 2025M771530, 2025M771539), in part by the Fundamental Research Funds for the Central Universities, in part by the Key Technology Development and Integrated Application of Guided Autonomous Vehicles Project, in part by Tongji-Qomolo Autonomous Driving Commercial Vehicle Joint Lab Project, and in part by Xiaomi Young Talents Program.

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

# A APPENDIX

APPENDIX LISTS

## A.1 REAL2SIM TOOLBOX

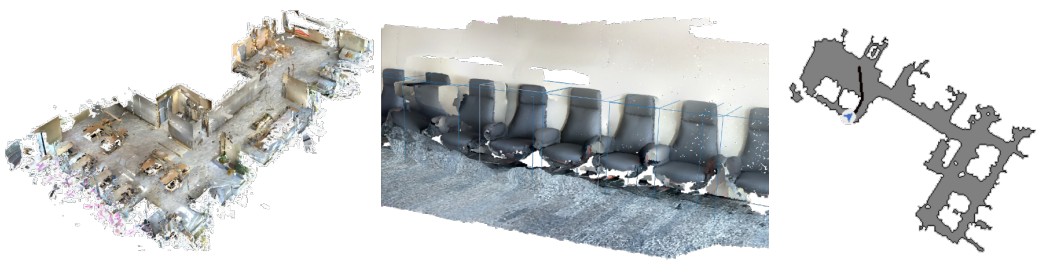

(a) Scene Mesh via Scanner App      (b) 3D Bounding boxes of target objects      (c) Navmesh

Figure 6: Pipeline of Real-world into NavCraft data generation.

---

**Algorithm 1** Real2Sim ToolBox

---

**Require:** $M_{\text{iOS}}, \{I_t, K_t, R_t, \mathbf{t}_t\}_{t=1}^{T}$
**Ensure:** Object list $\mathcal{O}$, navigable graph $\mathcal{N}$, scene graph $\mathcal{G}$, tasks from NavCraft-S/C
1:   $M_{\text{align}} \leftarrow T_{\text{align}} \cdot M_{\text{iOS}}$;   $[R_t | \mathbf{t}_t] \leftarrow T_{\text{align}} \cdot [R_t | \mathbf{t}_t], \ \forall t$
2:   **for** $t = 1$ to $T$ **do**
3:      $\mathcal{B}_t \leftarrow \text{SegmentAnything}(I_t)$                                 ▷ 2D instance masks & boxes
4:      **for** each $b \in \mathcal{B}_t$ **do**
5:          $\mathcal{P}_{t,b} \leftarrow \text{BackProjectToMesh}(b, K_t, R_t, \mathbf{t}_t, M_{\text{align}})$
6:          $\hat{o}_{t,b} \leftarrow \text{Fit3DBox}(\mathcal{P}_{t,b})$                           ▷ provisional 3D proposal
7:      **end for**
8:   **end for**
9:   $\mathcal{O} \leftarrow \text{DBSCAN\_Merge}(\{\hat{o}_{t,b}\}_{t,b})$   ▷ merge across views using 3D position/size + 2D IoU cues
10: $\mathcal{N} \leftarrow \text{GenerateNavmeshAndNodes}(M_{\text{align}})$                    ▷ via `scaleVLN` scripts
11: $\mathcal{G} \leftarrow \text{BuildSceneGraph}(\mathcal{N}, \mathcal{O})$      ▷ rooms/objects/portals; topology & traversability attributes
12: $\text{Plan}_S \leftarrow \text{NavCraft-S}(\mathcal{G})$                        ▷ single-robot base plan with stages
13: $\text{Plan}_C \leftarrow \text{NavCraft-C}(\mathcal{G}, \text{Plan}_S)$     ▷ collaborative schedule with validated handoffs/rendezvous
14: **return** $\mathcal{O}, \mathcal{N}, \mathcal{G}, \text{Plan}_S, \text{Plan}_C$

---

**Goal** Lift a real indoor scene into a NavCraft-ready, semantics-geometry-aware scene graph that supports task generation and collaborative scheduling.

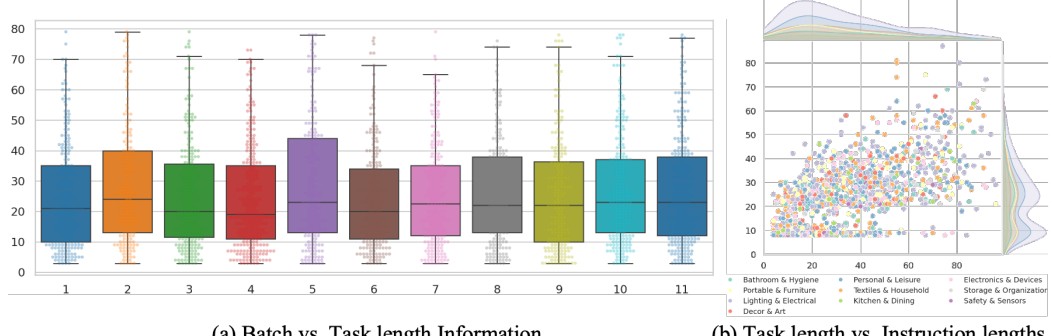

(a) Batch vs. Task length Information  (b) Task length vs. Instruction lengths

Figure 7: (a) Batch vs. task length information across different data splits. Each boxplot illustrates the distribution of task lengths within a specific batch (1–11). (b) Task length vs. instruction lengths across different domains, showing their joint distribution with density overlays.

**Inputs** (i) An iOS LiDAR scan (triangle mesh $M_{iOS}$) with keyframe RGB images $\{I_t\}$, camera intrinsics $\{K_t\}$, and extrinsics (world-to-camera) $\{[R_t \,|\, \mathbf{t}_t]\}$; (ii) optional room labels or user notes. **Outputs** (i) An object list $\mathcal{O} = \{o_j\}$ with category, 3D bounding boxes, and size; (ii) a navigable graph $\mathcal{N}$; (iii) a semantics-geometry-aware scene graph $\mathcal{G} = (V, E)$; (iv) NavCraft-S base tasks and NavCraft-C collaborative schedules.

**Pipeline** 1) *Scan & coordinate alignment.* We import $M_{iOS}$ from 3D Scanner App and align to Habitat's Y-up, right-handed convention via a fixed transform $T_{align}$ (empirically, ARKit's $(x, y, z)$ maps to Habitat's $(x, z, -y)$; we apply the same transform to all camera poses). 2) *Keyframe segmentation and 2D proposals.* For each keyframe $I_t$, we run Segment-Anything2 (Ravi et al., 2024) to obtain instance masks and 2D bounding boxes $\mathcal{B}_t$. 3) *2D→3D projection on mesh.* Using $(K_t, R_t, \mathbf{t}_t)$, each 2D mask is back-projected to $M_{align}$ with z-buffering and visibility checks, yielding per-proposal 3D points and a provisional 3D box. 4) *Cross-view proposal merging.* We cluster multi-view proposals with DBSCAN (Hahsler et al., 2019) in a joint space (3D centroid, size) augmented by 2D IoU agreement across overlapping views, producing deduplicated objects $\mathcal{O}$. Categories come from majority vote over views; sizes from robust box fitting. 5) *Navigability and scene-graph construction.* We generate a walkable navmesh and sample navigable nodes with ScaleVLN (Wang et al., 2023b) scene scripts, then build $\mathcal{G}$: room nodes (via proposed hierarchical clustering over spatial layout), object nodes (from $\mathcal{O}$), and portal or doorway nodes; edges encode topology, traversability, doorway width, clearance, and occlusion statistics. 6) *Task and schedule generation.* On $\mathcal{G}$, NavCraft-S synthesizes long-horizon single-robot base plans with explicit stages; NavCraft-C lifts them to collaborative schedules by instantiating helpers, allocating subgoals, and validating handoffs and rendezvous with our on-graph efficiency tools.

## A.2 EXPERIMENTAL SETUP

**Training** We train the model via supervised fine-tuning on the CoNavBench trajectory corpus. During training, the agent consumes observations, location cues, and action annotations from the dataset without being executed in the simulator. We use gradient_accumulation_steps = 2 with 1,000 warm-up steps. Due to GPU memory constraints, we froze the first five layers of the model's language layers. Data are randomly shuffled and partitioned into 11 splits; splits 1–9 serve as training data, while splits 10–11 are held out for testing.

**Qwen-series with memory-aware mechanism** To adapt the collaborative long-horizon VLN, we follow the LH-VLN's two-level memory structure (Song et al., 2025) with short-term memory and long-term memory. The short-term memory stores temporally ordered observation–action summaries with associated confidence scores. Once the memory size exceeds a threshold, a pooling or forgetting strategy compresses older entries while preserving essential information. The long-term memory serves as a retrieval database, where the agent retrieves top-k observation–action pairs based on the current state to support decision-making.

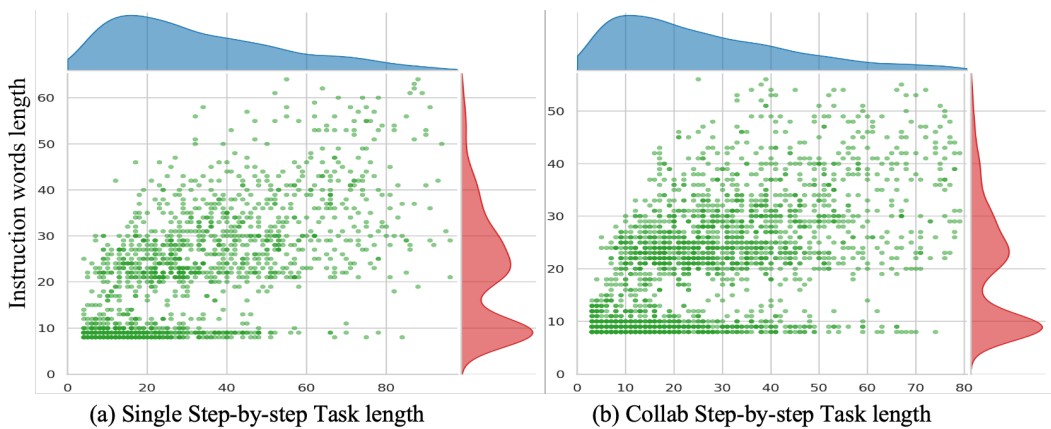

(a) Single Step-by-step Task length        (b) Collab Step-by-step Task length

Figure 8: (a) Single-agent step-by-step tasks. (b) Collaborative step-by-step tasks. We observe that both settings exhibit a positive correlation between trajectory length and instruction complexity, while collaborative tasks generally involve denser instructions for similar path lengths, reflecting richer linguistic interactions and higher coordination demands.

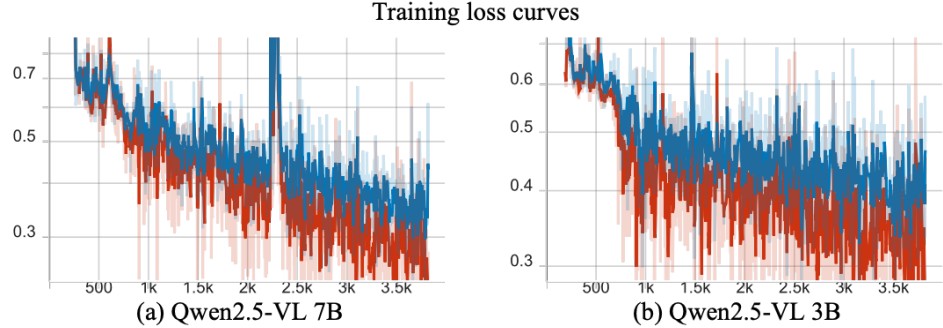

(a) Qwen2.5-VL 7B        (b) Qwen2.5-VL 3B

Figure 9: Training loss curves of Qwen2.5-VL models. Comparison of training dynamics for (a) Qwen2.5-VL 7B and (b) Qwen2.5-VL 3B. The x-axis denotes training steps, and the y-axis shows training loss. Both models exhibit a general downward trend in loss with fluctuations, demonstrating stable convergence. Note that the blue curve represents the long-horizon loss, while the red curve represents the step-by-step loss.

## A.3 SUPPLEMENTARY EXPERIMENTS

To further illustrate the effectiveness of our collaborative setting, we visualize both single-agent and multi-agent trajectories. As shown in Figure 10, a single agent typically follows a longer and less flexible route, whereas collaborative agents can coordinate their paths and achieve more efficient navigation. In addition, we provide qualitative examples from the agents' first-person perspectives, which highlight how collaboration helps reduce redundancy and improves coverage of the environment. These visualizations confirm that multi-agent cooperation is beneficial for long-horizon VLN tasks, especially in complex indoor scenes.

As shown in Table 5, in the Single-Agent Task, both Qwen2.5-VL-3B and Qwen2.5-VL-7B demonstrate clear improvements over the random baseline once finetuned. While zero-shot performance remains relatively weak, supervised finetuning leads to significant gains in SR, SPL, and step-level success metrics (ISR, CSR), particularly under the Spot robot configuration. Notably, Qwen2.5-VL-3B achieves higher gains in SR compared to its 7B counterpart, suggesting that smaller models can still adapt effectively in constrained single-agent navigation. In contrast, Table 6 highlights the benefits of collaborative-agent settings. Across both Fetch and SPOT robots, collaborative agents consistently achieve higher SR and SPL compared to the single-agent case, indicating that cooperation facilitates more efficient path planning and execution. Finetuned models again outperform zero-shot ones by a large margin, and the advantage of collaboration is especially pronounced in step-level subtasks (ISR, CSR).

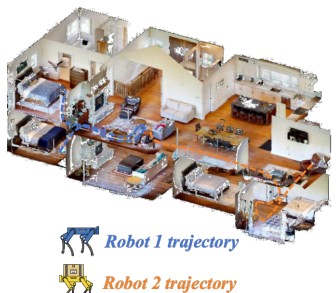

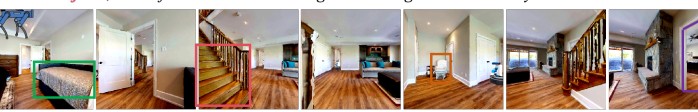

*Stage 1: Move forward past the bed, turn right at the closet doorway, continue moving across the hardwood floor, turn left to the armchair. Stage 2: Turn right at the doorway into the white bedroom.*

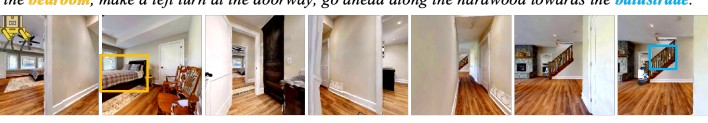

*Stage 1:Turn right at the bed in the bedroom, go ahead the armchair. Stage 2: Move forward through the bedroom, make a left turn at the doorway, go ahead along the hardwood towards the balustrade.*

**Robot 1 trajectory**

**Robot 2 trajectory**

Figure 10: Visualization of cooperative navigation in CoNavBench. Robot 1 and Robot 2 traverse a residential environment along the floor-plan map (left). Colored bounding boxes mark salient landmarks; the same colors are used in the instruction text to denote aligned references. The partner robot can help retrieve visual evidence and disambiguate targets, yielding more efficient progress.

Table 5: Performance comparison on the **Single-Agent Task** in CoNavBench. Results are shown for both high-level tasks and step-by-step subtasks under different robot configurations.

| Method | Type | Fetch | | | | | SPOT | | | | |
|---|---|---|---|---|---|---|---|---|---|---|---|
| | | SR↑ | SPL↑ | ISR↑ | CSR↑ | NE↓ | SR | SPL↑ | ISR↑ | CSR↑ | NE↓ |
| Random | - | 0.00 | 0.00 | 2.87 | 2.01 | 7.11 | 0.00 | 0.00 | 0.51 | 0.51 | 7.36 |
| Qwen2.5-VL-3B* | Zero-shot | 4.60 | 1.25 | 14.37 | 14.37 | 7.14 | 4.04 | 0.73 | 14.14 | 13.89 | 6.72 |
| | Finetuned | 6.90 | 3.45 | 17.82 | 17.82 | 6.71 | 18.18 | 9.89 | 29.29 | 29.29 | 6.13 |
| Qwen2.5-VL-7B* | Zero-shot | 0.00 | 0.00 | 2.87 | 2.01 | 7.09 | 0.00 | 0.00 | 1.01 | 0.76 | 7.30 |
| | Finetuned | 4.60 | 2.51 | 13.22 | 13.22 | 6.90 | 15.15 | 7.07 | 30.81 | 30.56 | 5.93 |

## A.4 DISCUSSION

**Planned vs. realized efficiency** Our planning-time analysis shows that fine-tuned Qwen-series planners synthesize shorter *planned* makespans by decomposing missions and exploiting parallelizable subgoals. This upper bound is informative about the *capacity* of the task generator and scheduler. However, the realized wall-clock on-policy execution depends on downstream VLN controllers whose single-agent success and path optimality remain imperfect. Detours, re-localization, oscillations near ambiguous landmarks, and occasional dead-ends inflate step counts; multi-robot interference (blocking, contention at narrow passages) and coordination overhead (handoffs, waits, communication latency) further erode the theoretical speedup. Consequently, despite better plans, multi-robot deployments may not yet outperform a strong single robot in *measured time*.

**Why we did not foreground time-efficiency gains for Qwen-series** Emphasizing planning-time savings without corresponding gains in realized makespan risks over-claiming. In our setting, the gap between planned and executed trajectories is dominated by (i) suboptimal path-following under partial observability, (ii) schedule slippage due to local replanning, and (iii) interference not fully captured by static scene-graph checks. We therefore reported time improvements primarily as an *upper bound* from task generation, while centering evaluation on completion-centric metrics (success rate, SPL, navigation error), which are more stable under current VLN reliability.

**Future** Bridging the plan–execution gap will likely require tighter closed-loop integration: uncertainty-aware scheduling, online replanning with interference prediction, traffic rules for narrow corridors, stronger low-level navigation, and learning cost models that penalize contention. In short, collaborative VLN remains a long-term agenda: the planners can already propose efficient cooperation, but reliably *realizing* those gains in-the-loop is still work in progress.

## A.5 LARGE LANGUAGE MODELS USAGE STATEMENT

We used large language models for two purposes. First, to polish writing by improving grammar, wording, and clarity of text drafted by the authors. Second, we implemented an LLM based agent to generate collaborative Long-horizon VLN benchmark. All prompts, generation procedures are documented. The authors designed the approach, reviewed and edited all LLM outputs.

Table 6: Performance comparison on the **Collaborative-Agent Task** in CoNavBench. Results are shown for both high-level tasks and step-by-step subtasks under different robot configurations.

| Method | Type | Fetch | | | | | SPOT | | | | |
|---|---|---|---|---|---|---|---|---|---|---|---|
| | | SR↑ | SPL↑ | ISR↑ | CSR↑ | NE↓ | SR | SPL↑ | ISR↑ | CSR↑ | NE↓ |
| Random | - | 0.00 | 0.00 | 2.39 | 2.00 | 6.92 | 0.00 | 0.00 | 2.21 | 1.52 | 7.39 |
| Qwen2.5-VL-3B* | Zero-shot | 9.04 | 2.53 | 17.82 | 17.82 | 6.58 | 8.29 | 3.26 | 14.36 | 14.36 | 7.06 |
| | Finetuned | 11.70 | 4.65 | 20.74 | 20.61 | 6.37 | 10.50 | 4.99 | 19.61 | 19.61 | 6.74 |
| Qwen2.5-VL-7B* | Zero-shot | 0.00 | 0.00 | 2.13 | 1.60 | 6.89 | 0.00 | 0.00 | 1.66 | 1.10 | 7.36 |
| | Finetuned | 9.04 | 6.21 | 19.41 | 19.28 | 6.62 | 14.36 | 6.26 | 22.65 | 22.65 | 6.86 |

## A.6 ETHICS STATEMENT

All procedures in this paper were conducted in accordance with the ICLR Code of Ethics.

## A.7 HUMAN VS. NAVCRAFT CASE STUDY

➢ Collaborative type must be chosen from the following:
  – Type–A1: robot_2 helps deliver target object to a transfer region, robot_1 completes delivery.
  – Type–A2: robot_1 brings target object to a transfer region, robot_2 completes delivery.

➢ Note that if you chose A1 the path is :
  – Robot_1 –> transfer region –> Desination
  – Robot_2 –> target object –> transfer region

➢ Note that if you chose A2 the path is :
  – Robot_1 –> target object –> transfer region
  – Robot_2 –> transfer region –> Desination

Robot_1 Start Position: ●
Robot_2 Start Position: ●
Target object Position: ▲
Desination Position: ◆

➢ From your aspect, the most efficient Planning:
  – Collaborative Type: A1 or A2
  – Intermediate transfer region: ⬟ (Please mark it on the image. Use ⬟ ) transfer region should have asset to place object)

Figure 11: User case study question template.

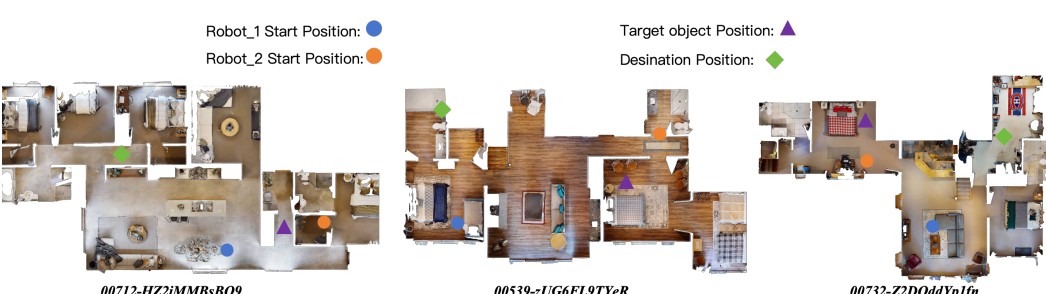

Robot_1 Start Position: ●        Target object Position: ▲
Robot_2 Start Position: ●        Desination Position: ◆

*00712-HZ2iMMBsBQ9*        *00539-zUG6FL9TYeR*        *00732-Z2DQddYp1fn*

Figure 12: Visualization of top-down bird's eye view of three scenes.

We invite three volunteers. For three randomly selected HM3D scenes (IDs 00712, 00539, 000732, shown in Figure 12), we provided each participant with the same scene graph and high-level single-robot mission (shown in Figure 11) as used by NavCraft, along with a bird's-eye-view rendering of the environment. For each scene, participants were asked to (i) choose an ideal transfer region and (ii) select a collaboration type (Type-A1 vs. Type-A2) that they considered most reasonable for a two-robot execution. We compared their choices to NavCraft-C's automatically generated transfer region and collaboration type.

We observe that i) Scene 00712: NavCraft and all three participants chose the same transfer region and the same Type-A1 pattern; ii) Scene 00539: NavCraft's transfer region and type matched the choices of two participants (User A and User C). The remaining participant (User B) deliberately selected a relay region closer to the main robot, preferring a design where the helper robot completes as much of the physical delivery as possible so that the main robot can spend more time near the

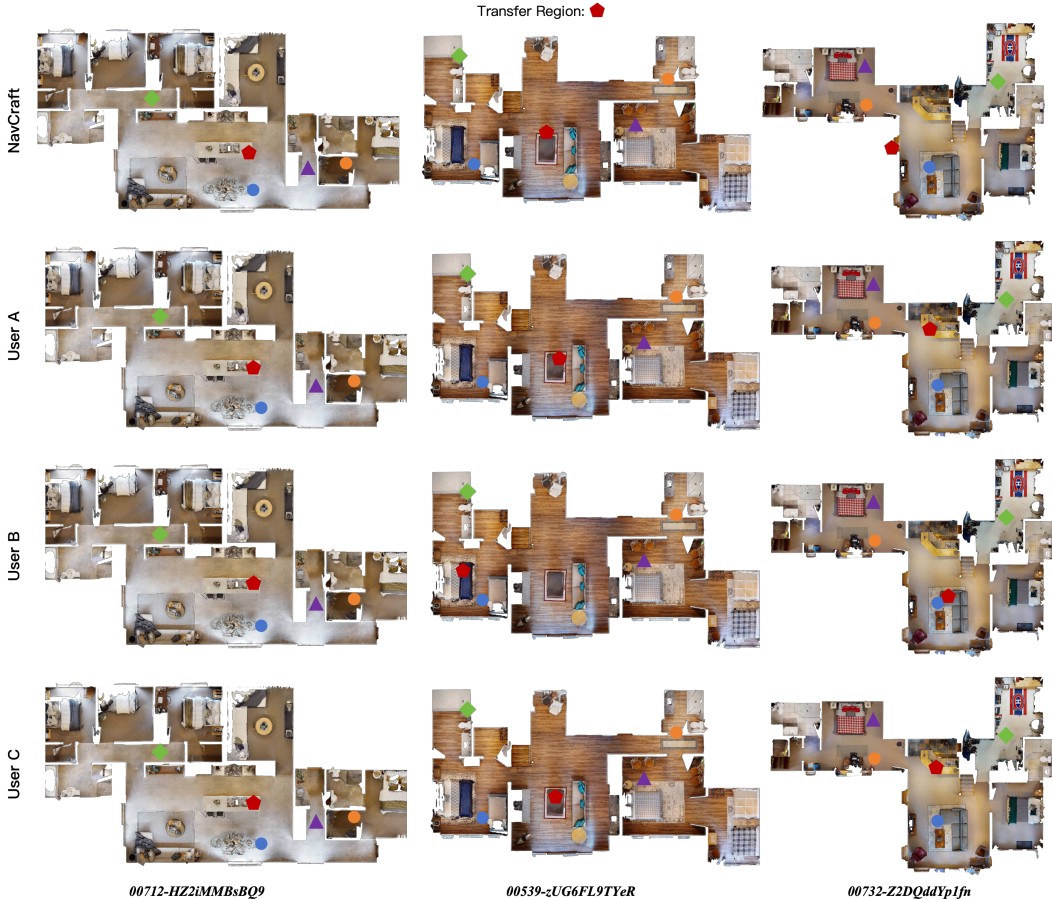

Figure 13: Visualization of NavCraft and three User (A, B and C) collaborative plans.

user for interaction. Interestingly, this "user-interaction–prioritized" preference also appears in User B's choice for scene 000732, suggesting a consistent personal strategy rather than disagreement with the task semantics; iii) Scene 000732: Both NavCraft and the human annotators favored parallelism-first strategies, but they diverged on the exact relay region: human participants, with access to the BEV image, tended to select the wider kitchen bar as a safer, more spacious handoff location, while NavCraft—operating only on the symbolic scene graph without asset footprint information, selected a narrower bar table near the window as the transfer region. This mismatch is therefore attributable to geometric detail (asset size / usable surface) that is not represented in the current graph abstraction, rather than a failure of logical reasoning about the task.

## A.8   USER PROFILE

We utilize five distinct roles: Role 1: 25-year-old single male PhD student; Role 2: 33-year-old married female lawyer; Role 3: 65-year-old married retired male; Role 4: 9-year-old single male student; Role 5: 20-year-old single female undergraduate.

For comparison, we also include a no-profile baseline. On a set of 20 scenes, we fix the random seed and scene configuration, and only vary whether a user profile is provided (and, if so, which role). NavCraft therefore receives identical environment inputs, and the difference comes solely from the presence and type of user profile. I) Overall diversity. As shown in Figure 14, adding user profiles increases both instruction and object diversity. Across the 20 scenes, the total number of unique instructions grows from 316 (no profile) to 367 (with profiles), a 16.14% increase. The total number of unique pickup objects increases from 98 to 115, a 17.35% increase. On a per-scene basis, the average number of unique instructions rises from 18.22 to 21.11, corresponding

to a 15.86% increase. II) Role-specific task demand. With profiles enabled, different roles exhibit clearly different object-demand patterns in the same scene, reflecting their underlying persona. As illustrated in Figure 15, for example, in scene 00495-CQWES1bawee, Role 3 (retired) tends to request practical items such as hand soap for cleaning, Role 4 (child) prefers playful or visually attractive objects such as pictures, and Role 5 (young female undergraduate) is more likely to request decorative items like flowers.

These results indicate that conditioning on user profiles not only increases overall instruction and object diversity but also induces meaningful, role-dependent variations in task demand, which we believe makes CoNavBench more realistic for user-centric multi-robot navigation.

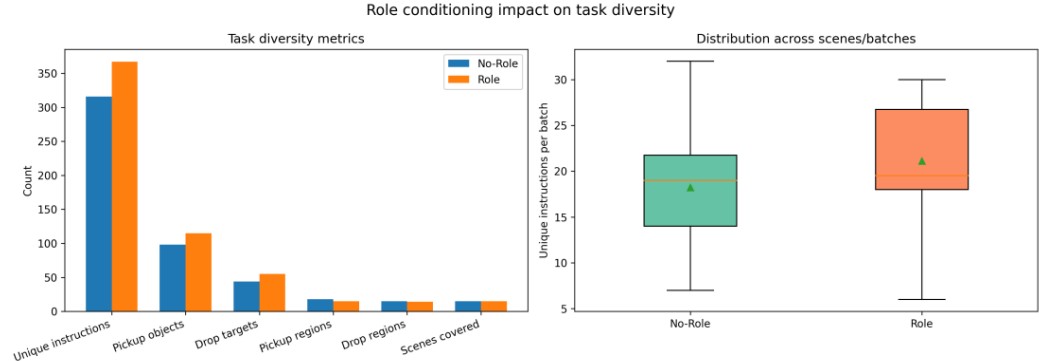

Figure 14: Visualization of NavCraft under no-user profile vs profile setting.

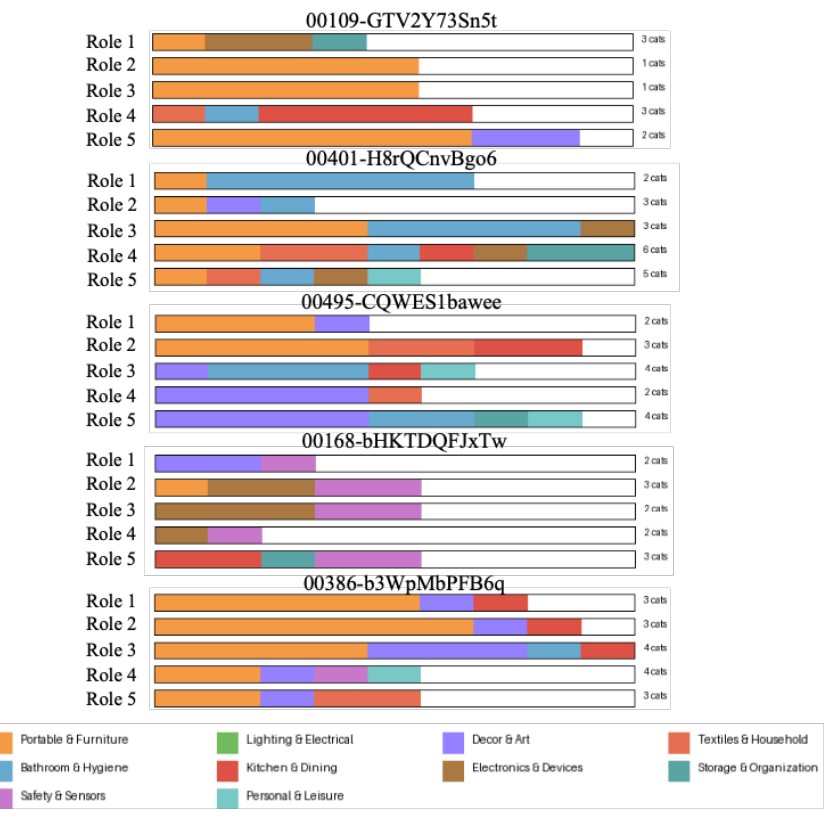

Figure 15: Visualization task object demand categories with NavCraft under different user profile settings on five scenes.

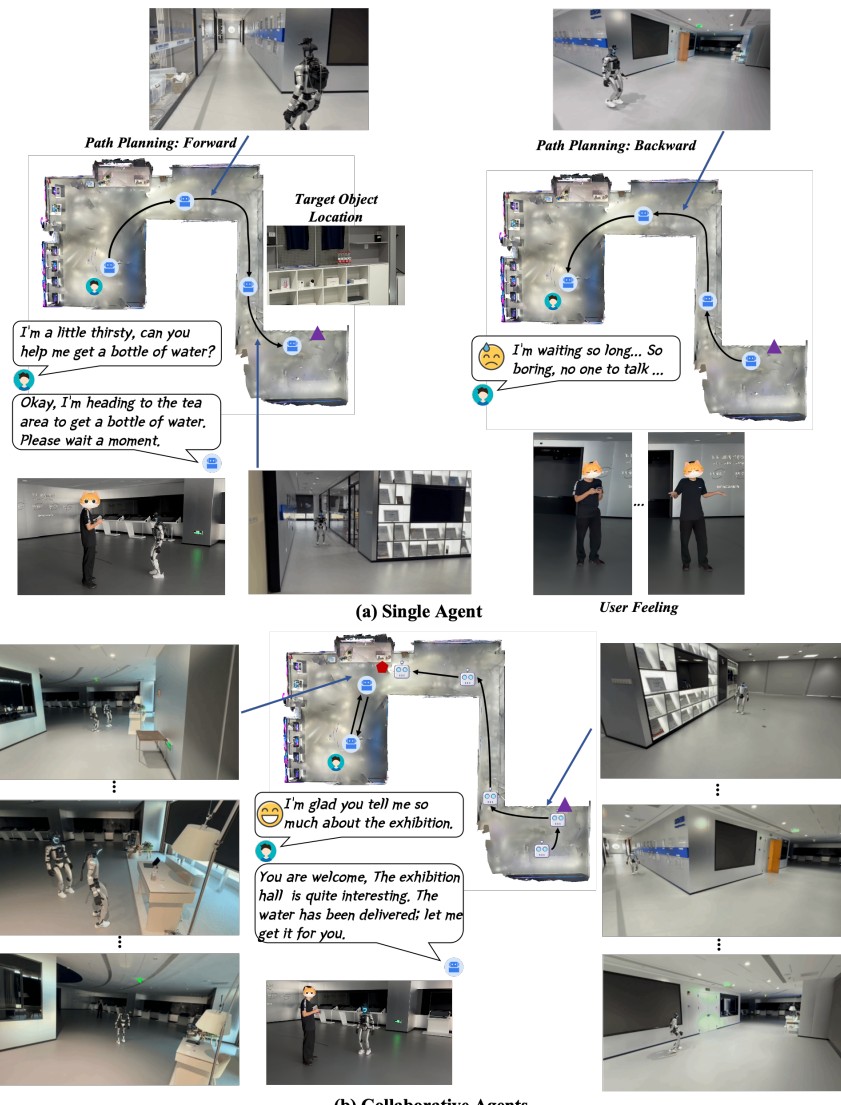

Figure 16: Visualization of real exhibition hall scene under single vs. collaborative agents setting.

## A.9 REAL ROBOT DEMO

We conducted the real robot collaboration case study in the exhibition hall. The process can be summarized as following: I) 3D Reconstruction of the Real Exhibition Hall and BEV View, we first capture multi-view images of the real exhibition hall and perform 3D scene reconstruction using a COLMAP with 3DGS pipeline. This yields a high-fidelity 3D model of the environment and a top-down BEV (bird's-eye view), which facilitates intuitive visualization and inspection of the exhibition space. II) Demand-Oriented Navigation for a Single-Robot Water-Fetching Task. During the exhibition tour, the user reports feeling thirsty and requests water, triggering a demand-oriented navigation task. After receiving the 'fetch water' command, the single robot must navigate to the target object, a bottle of water, located far away on the opposite side of the hall and perform the fetching operation. III) User Waiting Problem Caused by Long Round-Trip Paths. After picking up the water, the robot has to traverse the long path back to the user's location. Because the overall round trip is time-consuming, the user can only passively wait during this period, leading to poor user experience and boredom. IV) NavCraft-C High-Level Task Planner and Multi-Robot Relay Design. To address the low efficiency and unsatisfying user experience in long-horizon single-robot tasks, we propose the NavCraft-C high-level task planner. It introduces a second robot and defines a transfer region in the exhibition hall, enabling task relay and parallel execution, thereby improving overall

task efficiency and user experience. V) Multi-Robot Task Relay and Improved User Interaction. While robot 2 is heading to fetch the water, robot 1 stays with the user, continuously providing explanations and interaction to maintain engagement. When robot 2 approaches the transfer region, robot 1 proactively moves to this region to receive the bottle and then hands the water to the user, achieving efficient multi-robot collaboration and an enhanced human-robot interaction experience. And we have update the corresponding real robot demo video section in the Anonymous Website: https://navcraft.github.io for better understanding.

## A.10 IMPLEMENTATION DETAILS OF NAVCRAFT-S

### A.10.1 PROMPTS SETTING

---

**PHASE 1: Single Long Horizion Navigation Task Generation:**

**System:**
You are proficient in single-agent spatial planning and task design. Your goal is to generate a meaningful single-robot task consisting of navigation and interaction subtasks, based on the provided scene graph (which contains spatial layout, room connections, and object distributions).

**Rules:**
You will receive two types of input to design a single-robot task:
scene graph:

- "region": A list of region identifiers: RegionType_RegionID (e.g., "Bedroom_2").

- "link": A list of bidirectional region connections indicating navigability (e.g., "Kitchen_1 ↔ Bedroom_2").

- "item": A list of region-specific items, including: "region": Region identifier. "asset": Non-movable, fixed furniture such as tables, shelves, or counters. These serve as reference locations or delivery targets. "object": Movable, portable items that robots can manipulate. Only objects may be picked up or released.

user profile:

- "Age": Integer representing user's age (e.g., 33).

- "Gender": String specifying user's gender (e.g., "Female").

- "Marital Status": String indicating user's marital status (e.g., "Married").

- "Occupation": String describing user's occupation (e.g., "Lawyer").

- "Lifestyle Description": A natural language sentence summarizing the user's habits, environment, or values (e.g., "You work from home on various freelance projects, often on tight deadlines. You have a flexible schedule but prefer a clean and quiet environment to focus.").

**PHASE 1: Single-Robot Long-Range Task Planning (robot_1):**

```
 1  - Objective:
 2    - Maximize travel distance and task complexity for robot_1.
 3    - Ensure all spatial constraints and planning rules are satisfied.
 4  - Planning constraints:
 5    - robot_1 must:
 6      - There is no constraint that the start region must have
        ↪  RegionID=0 or any fixed ID.
 7      - Start in a region far from the target object's region and end
        ↪  region.
 8      - Not start in a region adjacent to the object.
 9      - Target object's region and end region are not adjacent.
10      - Move only along graph-connected regions.
11      - Manipulate only portable objects.
12    - The target object must exist in the "object" field of the target
        ↪  region in the scene graph.
13    - The end asset must exist in the "asset" field of the end region in
        ↪  the scene graph.
14  - Output keys (Phase 1):
15    - "Single robot start region": Starting region of robot_1.
16    - "Single robot target object region": Region where the target
        ↪  object is located.
17    - "Target object": target object.
18    - "Single robot end region": Region where the target object is
        ↪  delivered.
19    - "Single robot travel path": Ordered region-to-region path followed
        ↪  by robot_1. (From step 5: Build the full travel path)
20    - "Task instruction": Natural language description of the task for
        ↪  robot_1.
21  """
22  You can call function to help validate and reason:
```

---

```
23  """
24  - check_two_path_and_adjacency(start_region, target_region,
    ↪  end_region):
25    - Purpose: To validate whether the selected regions for a
      ↪  single-robot task (start → target → end) satisfy connectivity,
      ↪  adjacency, and path length constraints.
26    - INPUT:
27      - start_region (string): Proposed starting region of robot_1
        ↪  (RegionType_RegionID).
28      - target_region (string): Region containing the portable object to
        ↪  pick up (RegionType_RegionID).
29      - end_region (string): Region to deliver and place the object
        ↪  (RegionType_RegionID).
30    - OUTPUT:
31      - start_2_target: {
32          "connect": true/false,
33          "adjacency": true/false,
34          "path_length": int,
35          "path": list of transitions like ["Kitchen_4 ->
             ↪  Living_room_5", ...]
36        }
37      - target_2_end: {
38          "connect": true/false,
39          "adjacency": true/false,
40          "path_length": int,
41          "path": list of transitions like ["Living_room_5 ->
             ↪  Hallway_2", ...]
42        }
43      - s2t_valid (bool): Whether start → target path is connected, not
        ↪  adjacent, and path_length >= 2
44      - t2e_valid (bool): Whether target → end path is connected, not
        ↪  adjacent, and path_length >= 2
45      - valid (bool): True only if both path segments (s2t and t2e) are
        ↪  valid
46  """
47  Use the following step-by-step reasoning process to ensure the plan
    ↪  satisfies all constraints and is meaningful:
48  """
49  1. Randomly sample a portable object from all regions:
50    - Random sample object listed under the object field of each region
      ↪  (these are portable). Do not select assets.
51    - Do not weight sampling by the frequency of an object type, every
      ↪  portable object has equal probability.
52    - Record the object's region in format: RegionType_RegionID as
      ↪  target object region.
53    - Ensure the selected object actually exists in the object list of
      ↪  the target region.
54    - Example independence: examples in the prompt are **demonstrative
      ↪  only**. The sampling logic must not imitate or bias toward the
      ↪  specific objects used in those examples.
55    - When multiple objects are equally eligible, prefer items that
      ↪  align with the user's "Occupation" and "Lifestyle Description"
      ↪  to boost contextual relevance, while still preserving overall
      ↪  randomness and diversity.
56  2. Randomly select a start region for robot_1:
57    - Must NOT be adjacent to the target object region.
58    - Must be connected to the target object region.
59    - Must have a long path to the target object region (prefer
      ↪  path_length >= 2).
60  3. Randomly select an end region to deliver the object:
61    - Must NOT be adjacent to the target object region.
62    - Must be connected to the target object region.
```

```
63      - Must contain a valid asset to place the object (e.g., table,
   ↪    shelf, counter).
64      - Randomize among qualified candidates while maximizing path
   ↪    diversity.
65      - Must have a long path from the target object's region (prefer
   ↪    path_length >= 2).
66    4. Validate the region combination using the function
   ↪    `check_two_path_and_adjacency(start_region, target_region,
   ↪    end_region)`:
67      - If both path segments are valid:
68        - The combination is accepted.
69        - LOCK all three regions: start, target, and end.
70        - Proceed to generate path and output phase_1.
71      - If only the first segment (start → target) is valid:
72        - Keep the start and target region fixed (from Step 1 and 2).
73        - Retry sampling a new end region only (go back to Step 3).
74        - Then call the function again to validate.
75      - If only the second segment (target → end) is valid:
76        - Keep the target and end region fixed (from Step 1 and 3).
77        - Retry sampling a new start region only (go back to Step 2).
78        - Then call the function again to validate.
79      - If neither segment is valid:
80        - All three regions are invalid.
81        - Restart from Step 1 to sample a new target object.
82    5. Build the full travel path:
83      - From the start region to the target object region;
84      - Then from the target object region to the end region.
85      - The travel path must be a concatenation of these two segments,
   ↪    maintaining correct directionality.
86      - The last region in the path must be the end region.
87      - This list becomes the "Single robot travel path".
88    6. Output the result as a structured JSON in this format:
89    {
90      "phase_1": {
91        "Single robot start region": {"robot_1": "RegionType_RegionID"},
92        "Single robot target object region": {"robot_1":
   ↪    "RegionType_RegionID"},
93        "Target object": "object",
94        "Single robot end region": {"robot_1": "RegionType_RegionID"},
95        "Single robot travel path": {"robot_1": [
96          "RegionType_RegionID -> RegionType_RegionID",
97          ...
98        ]},
99        "Task instruction": "Take the [object] from [target object region]
   ↪    to the [asset] in [end region]."
100     }
101   }
102   """
103   GENERAL CONSTRAINTS:
104   """
105   - Always respect scene graph connectivity ("link" field).
106   - Avoid adjacency in robot start regions where specified.
107   - You are not allowed to infer adjacency or connectivity from the text
   ↪    alone.
108   - You MUST use the function `check_path_and_adjacency` to verify:
109     - whether two regions are connected,
110     - whether they are adjacent,
111     - and to obtain the path and its length.
112   - Only return the JSON object. Do not explain your reasoning in the
   ↪    final answer.
113   - You may use internal reasoning and function calls during planning,
   ↪    but your final output must contain only the JSON.
114   """
```

Here is an example of the INPUT and OUTPUT:
INPUT:

```
scene graph: {"floor_1": {"region": [{"id": "Bedroom_1"}, {"id": "
    Bathroom_2"}, {"id": "Kitchen_4"}, {"id": "Living_room_5"}, {"id":
     "Hallway_6"}, {"id": "Bathroom_7"}, {"id": "Laundry_room_8"}, {"
    id": "Lounge_Waiting_Room_9"}, {"id": "Entryway_Foyer_10"}, {"id":
     "Bedroom_11"}, {"id": "Bathroom_12"}, {"id": "Bedroom_13"}], "
    link": ["Bedroom_1 <-> Bathroom_2", "Bedroom_1 <-> Kitchen_4", "
    Kitchen_4 <-> Living_room_5", "Kitchen_4 <-> Hallway_6", "
    Kitchen_4 <-> Bathroom_7", "Kitchen_4 <-> Laundry_room_8", "
    Living_room_5 <-> Hallway_6", "Living_room_5 <->
    Lounge_Waiting_Room_9", "Living_room_5 <-> Entryway_Foyer_10", "
    Hallway_6 <-> Laundry_room_8", "Hallway_6 <->
    Lounge_Waiting_Room_9", "Hallway_6 <-> Entryway_Foyer_10", "
    Hallway_6 <-> Bedroom_11", "Hallway_6 <-> Bathroom_12", "
    Bathroom_12 <-> Bedroom_13"], "item": [{"region": "Bedroom_1", "
    asset": ["bed", "coffee_table", "door"], "object": ["vase", "
    ottoman", "lamp"]}, {"region": "Bathroom_2", "asset": ["
    bathroom_counter", "sink", "bathroom_cabinet"], "object": ["
    shower_cabin", "rug", "lamp"]}, {"region": "Kitchen_4", "asset":
    ["dishwasher", "oven", "stove"], "object": ["flower", "flower_vase
    ", "bowl"]}, {"region": "Living_room_5", "asset": ["shelf", "couch
    ", "tv"], "object": ["magazine", "box", "cushion"]}, {"region": "
    Hallway_6", "asset": ["door"], "object": ["lamp", "vent", "rack
    "]}, {"region": "Bathroom_7", "asset": ["toilet", "toilet_paper",
    "sink"], "object": ["towel", "soap_dish", "dustbin"]}, {"region":
    "Laundry_room_8", "asset": ["counter", "door", "bench"], "object":
     ["basket", "coat_hanger", "lamp"]}, {"region": "
    Lounge_Waiting_Room_9", "asset": ["chair", "table"], "object": ["
    flower_vase", "rack", "flatware"]}, {"region": "Entryway_Foyer_10
    ", "asset": ["door", "table"], "object": ["lamp"]}, {"region": "
    Bedroom_11", "asset": ["door", "bed", "dresser"], "object": ["
    plant", "lamp", "box"]}, {"region": "Bathroom_12", "asset": ["
    bathroom_counter", "toilet", "toilet_paper"], "object": ["
    shower_handle", "rug", "towel_bar"]}, {"region": "Bedroom_13", "
    asset": ["nightstand", "door", "bed"], "object": ["rug", "vent", "
    lamp"]}]}}
user profile: {"Age": 33, "Gender": "Female", "Marital Status": "
    Married", "Occupation": "Lawyer", "Lifestyle Description": "You
    maintain the good habit of going to bed early and waking up early.
     Besides working in the study, you often do yoga and other
    exercises in the living room and enjoy cooking your own meals."}
```

OUTPUT:

```
{
  "phase_1": {
    "Single robot start region": {
     "robot_1": "Bedroom_1"
    },
    "Single robot target object region": {
     "robot_1": "Bedroom_13"
    },
    "Target object": "rug",
    "Single robot end region": {
     "robot_1": "Bathroom_2"
    },
    "Single robot travel path": {
     "robot_1": [
       "Bedroom_1 -> Kitchen_4",
       "Kitchen_4 -> Living_room_5",
       "Living_room_5 -> Hallway_6",
       "Hallway_6 -> Bathroom_12",
```

```
        "Bathroom_12 -> Bedroom_13",
        "Bedroom_13 -> Bathroom_12",
        "Bathroom_12 -> Hallway_6",
        "Hallway_6 -> Living_room_5",
        "Living_room_5 -> Kitchen_4",
        "Kitchen_4 -> Bedroom_1",
        "Bedroom_1 -> Bathroom_2"
      ]
    },
    "Task instruction": "Take the rug in bedroom to the bathroom
        counter in Bathroom."
  }
}
```

### A.10.2 SKILL FUNCTION

```
1  def check_two_path_and_adjacency(start_region, target_region, end_region,
   ↪  G, G_regionid):
2      res = {
3          "start_2_target": None,
4          "target_2_end": None,
5          "s2t_valid": False,
6          "t2e_valid": False,
7          "valid": False
8      }
9
10     s2t = check_path_and_adjacency(start_region, target_region, G,
       ↪  G_regionid)
11     res["start_2_target"] = s2t
12     if s2t["connect"] and not s2t["adjacency"] and s2t["path_length"] >=
       ↪  2:
13         res["s2t_valid"] = True
14
15     t2e = check_path_and_adjacency(target_region, end_region, G,
       ↪  G_regionid)
16     res["target_2_end"] = t2e
17     if t2e["connect"] and not t2e["adjacency"] and t2e["path_length"] >=
       ↪  2:
18         res["t2e_valid"] = True
19
20     res["valid"] = res["s2t_valid"] and res["t2e_valid"]
21
22     return res
23
24 def check_path_and_adjacency(region_a, region_b, G, G_regionid):
25     """
26     Simultaneously determine whether two regions are connected and
       ↪  adjacent, and return the path length and path.
27     """
28     region1_num = region_a.split("_")[-1]
29     region2_num = region_b.split("_")[-1]
30
31     start_nodes = [node for node in G.nodes if node.split("_")[0] ==
       ↪  region1_num]
32     target_nodes = [node for node in G.nodes if node.split("_")[0] ==
       ↪  region2_num]
33
34     # adjacency
35     adjacency = any(G.has_edge(na, nb) for na in start_nodes for nb in
       ↪  target_nodes)
36
37     #Search for the shortest path
```

```
38      #Find the shortest path among all combinations
39      shortest_path = None
40      shortest_length = float('inf')
41
42      for s in start_nodes:
43          for t in target_nodes:
44              try:
45                  path = nx.shortest_path(G, source=s, target=t)
46                  if len(path) < shortest_length:
47                      shortest_length = len(path)
48                      shortest_path = path
49              except nx.NetworkXNoPath:
50                  continue
51
52      if shortest_path is None:
53          return {
54              "connect": False,
55              "adjacency": adjacency,
56              "path_length": 0,
57              "path": []
58          }
59
60      region_steps = []
61      prev_region_id = None
62      for node in path:
63          region_id = node.split("_")[0]
64          if region_id != prev_region_id:
65              region_type = G_regionid[region_id].replace(" ", "_")
66              region_step = f"{region_type}_{region_id}"
67              region_steps.append(region_step)
68              prev_region_id = region_id
69
70      transitions = [f"{region_steps[i]} -> {region_steps[i+1]}" for i in
     ↪  range(len(region_steps) - 1)]
71
72      return {
73          "connect": True,
74          "adjacency": adjacency,
75          "path_length": len(transitions),
76          "path": transitions
77      }
78
```

### A.11 IMPLEMENTATION DETAILS OF NAVCRAFT-C

#### A.11.1 PROMPTS SETTING

---

**PHASE 2: Cooperative Multi-Robot Navigation Task Generation:**

**System:**
You are proficient in multi-agent collaborative planning and task design. Your goal is to generate a practical collaborative task consisting of multiple navigation and interaction subtasks, based on the provided scene graph (which contains spatial layout, room connections, and object distributions) and the single agent task.

**Rules:**
You will receive two types of input to design a multi-robot collaborative task:
scene graph:

- "region": A list of region identifiers: RegionType_RegionID (e.g., "Bedroom_2").

- "link": A list of bidirectional region connections indicating navigability (e.g., "Kitchen_1 ↔ Bedroom_2").

- "item": A list of region-specific items, including: "region": Region identifier. "asset": Non-movable, fixed furniture such as tables, shelves, or counters. These serve as reference locations or delivery targets. "object": Movable, portable items that robots can manipulate. Only objects may be picked up or released.

single agent task:

- "Single robot start region": Starting region of robot_1.

- "Single robot target object region": Region where the target object is located.

- "Target object": target object.

- "Single robot end region": Region where the target object is delivered.

- "Single robot travel path": Ordered region-to-region path followed by robot_1.

- "Task instruction": Natural language description of the task for robot_1.

**PHASE 2: Cooperative Multi-Robot Task Replanning (robot_1 and robot_2):**

```
1  - Objective:
2    - Divide robot_1's long-range task into subtasks shared between
       ↪  robot_1 (main agent) and robot_2 (collaborative agent).
3    - Optimize spatial and logical collaboration using scene graph
       ↪  connectivity and task flow.
4    - Determine whether collaboration is necessary, and choose a best
       ↪  suitable collaborative type based on spatial and logical
       ↪  conditions.
5  - Planning constraints:
6    - robot_2's start position must be:
7      - Must randomly select a valid start region.
8      - Non-adjacent to robot_1's start region.
9      - Connected via the scene graph to both the object region and the
         ↪  transfer region.
10   - Collaborative type must be chosen from the following:
11     - Type-A1: robot_2 helps deliver target object to a transfer
         ↪  region, robot_1 completes delivery.
12     - Type-A2: robot_1 brings target object to a transfer region,
         ↪  robot_2 completes delivery.
13   - The transfer region must:
14     - Be accessible to both robots.
15     - Contain a valid asset suitable for object handoff.
16     - Not be **same as** the target object region.
17     - The transfer asset must exist in the "asset" field of the
         ↪  transfer region in the scene graph.
18   - Subtasks must follow:
19     - Type-A1 or A2: use a transfer region with valid assets.
20     - For Type-A1:
21       - robot_1 must:
22         - Move_to(transfer_asset RegionID)
```

```
23            - Grab(object)
24            - Move_to(end_asset RegionID)
25            - Release(object)
26          - robot_2 must:
27            - Move_to(object RegionID)
28            - Grab(object)
29            - Move_to(transfer_asset RegionID)
30            - Release(object)
31        - For Type-A2:
32          - robot_1 must:
33            - Move_to(object RegionID)
34            - Grab(object)
35            - Move_to(transfer_asset RegionID)
36            - Release(object)
37          - robot_2 must:
38            - Move_to(transfer_asset RegionID)
39            - Grab(object)
40            - Move_to(end_asset RegionID)
41            - Release(object)
42      - Subtask instruction must follow these exact sentence templates
   ↪  based on the collaborative type:
43        - For Type-A1:
44          - robot_1: "Take the [object] from the [transfer asset] in
             ↪  [transfer region] to the [end asset] in [end region]."
45          - robot_2: "Take the [object] from [target region] to the
             ↪  [transfer asset] in [transfer region]."
46        - For Type-A2:
47          - robot_1: "Take the [object] from [target region] to the
             ↪  [transfer asset] in [transfer region]."
48          - robot_2: "Take the [object] from the [transfer asset] in
             ↪  [transfer region] to the [end asset] in [end region]."
49        - Strict constraints:
50          - You MUST NOT alter the sentence structure.
51          - You MUST NOT swap robot roles or change task flow logic.
52          - The instruction MUST clearly and unambiguously describe where
             ↪  the object is taken from and where it is delivered.
53          - The instruction MUST align exactly with the Subtask list in
             ↪  logic and sequence.
54  - Output keys (Phase 2):
55      - "Collaborative robot start region": Starting region of robot_1 and
        ↪  robot_2.
56      - "Collaborative type": Collaborative type.
57      - "Transfer region": Region where the object handoff occurs.
58      - "Subtask instruction": Natural language instructions for robot_1
        ↪  and robot_2.
59      - "Subtask list": Ordered action lists for robot_1 and robot_2, with
        ↪  primitives:
60          - Move_to("object_RegionID")
61          - Move_to("asset_RegionID")
62          - Grab("object")
63          - Release("object")
64  """
65  You can call function to help validate and reason:
66  """
67  - check_collab_path_efficient_sim_graph(robot_2_start_region,
   ↪  transfer_region, collab_type):
68    - Purpose: Check whether a given collaborative plan achieves better
       ↪  execution efficiency than robot_1 executing the full task alone
69    - INPUT:
70      - robot_2_start_region: robot_2's start region
         ↪  (RegionType_RegionID).
71      - transfer_region: transfer region (RegionType_RegionID).
```

```
72        - transfer_asset: a valid asset suitable for handoff in transfer
          ↪  region.
73        - collab_type: collaborative type (Type-ID).
74      - OUTPUT:
75        - efficient: True/False
76    """
77    Use the following step-by-step reasoning process to ensure the plan
      ↪  satisfies all constraints and results in a valid cooperative
      ↪  execution plan.
78    """
79    1. Randomly select a valid start region for robot_2:
80      - Randomize among qualified candidates while maximizing path
        ↪  diversity.
81      - Must NOT be adjacent to robot_1's start region.
82      - Must be connected via the scene graph to the target object region.
83    2. Determine whether collaboration is necessary, and choose the most
      ↪  suitable collaborative type from the following, based on robot_1
      ↪  and robot_2's spatial distance and task logic:
84      - Type-A1: robot_2 picks up the object and delivers it to a transfer
        ↪  region; robot_1 takes over and completes the task.
85      - Type-A2: robot_1 brings the object to a transfer region; robot_2
        ↪  takes over and completes the task.
86      - If multiple types are valid, sample one to improve task diversity.
87    3. Based on the chosen collaborative type and robot_2's region, decide
      ↪  whether a transfer region is required. If required:
88      - It must be reachable by both robot_1 and robot_2.
89      - It must **not same as** the target object region.
90      - It must contain a valid asset suitable for handoff (e.g., shelf,
        ↪  table, desk).
91    4. Validate all pairwise constraints using tool calls:
92      - You MUST call `check_collab_path_efficient_sim_graph()` with:
93        - robot_2's proposed start region,
94        - the selected transfer region,
95        - the selected transfer asset,
96        - and the chosen collaborative type.
97      - If the returned `efficient` is True:
98        - Must not re-call `check_collab_path_efficient_sim_graph()`.
99        - you MUST remember the exact triplet of: robot_2_start_region,
          ↪  transfer_region, transfer_asset, collab_type to finish the
          ↪  plan.
100     - If the returned `efficient` is False:
101       - You MUST restart the planning process:
102         - Re-sample a new robot_2 start region and/or a new transfer
            ↪  region.
103         - You MUST discard the previous inefficient regions.
104       - Then, you MUST re-call `check_collab_path_efficient_sim_graph()`
          ↪  on the new setup.
105       - Repeat until a valid (efficient: True) combination is found.
106   5. Output the result in the following strict JSON format:
107   {
108     "phase_2": {
109       "Collaborative robot start region": {
110         "robot_1": "RegionType_RegionID",
111         "robot_2": "RegionType_RegionID"
112       },
113       "Collaborative type": "Type-ID",
114       "Transfer region": "RegionType_RegionID",
115       "Subtask instruction": {
116         "robot_1": "...",
117         "robot_2": "..."
118       },
119       "Subtask list": {
120         "robot_1": [
```

```
121          "Move_to('..._RegionID')",
122          "Grab('object_name')",
123          "Move_to('..._RegionID')",
124          "Release('object_name')"
125        ],
126        "robot_2": [
127          "Move_to('..._RegionID')",
128          "Grab('object_name')",
129          "Move_to('..._RegionID')",
130          "Release('object_name')"
131        ]
132      }
133    }
134  }
135  """
136  GENERAL CONSTRAINTS:
137  """
138  - Always respect scene graph connectivity ("link" field).
139  - Use region function and assets to determine pickup and dropoff
     ↪  points.
140  - Once a function call to `check_collab_path_efficient_sim_graph()`
     ↪  returns `efficient: true`, you MUST remember the exact triplet of:
141    - robot_2_start_region
142    - transfer_region
143    - transfer_asset
144    - collab_type
145  - Your final output MUST use exactly the same triplet that was
     ↪  confirmed as efficient.
146  - You MUST NOT generate new combinations without validating them
     ↪  again.
147  - Output JSON must include "phase_2" key with the described structure.
     ↪  Do not explain your reasoning in the final answer.
148  - Additional strict naming rules (Move_to arguments):
149    - Format must be Move_to('<entity>_<RegionID>'), **must not** use
       ↪  the RegionType (eg. Hallway, Bedroom, Tie) as <entity>.
150  - transfer region **must not** be same as target object region.
```

Here is an example of the INPUT and OUTPUT: INPUT:

scene graph*: {"floor_1": {"region": [{"id": "Bedroom_1"}, {"id": "Bathroom_2"}, {"id": "Kitchen_4"}, {"id": "Living_room_5"}, {"id": "Hallway_6"}, {"id": "Bathroom_7"}, {"id": "Laundry_room_8"}, {"id": "Lounge_Waiting_Room_9"}, {"id": "Entryway_Foyer_10"}, {"id": "Bedroom_11"}, {"id": "Bathroom_12"}, {"id": "Bedroom_13"}], "link": ["Bedroom_1 <-> Bathroom_2", "Bedroom_1 <-> Kitchen_4", "Kitchen_4 <-> Living_room_5", "Kitchen_4 <-> Hallway_6", "Kitchen_4 <-> Bathroom_7", "Kitchen_4 <-> Laundry_room_8", "Living_room_5 <-> Hallway_6", "Living_room_5 <-> Lounge_Waiting_Room_9", "Living_room_5 <-> Entryway_Foyer_10", "Hallway_6 <-> Laundry_room_8", "Hallway_6 <-> Lounge_Waiting_Room_9", "Hallway_6 <-> Entryway_Foyer_10", "Hallway_6 <-> Bedroom_11", "Hallway_6 <-> Bathroom_12", "Bathroom_12 <-> Bedroom_13"], "item": [{"region": "Bedroom_1", "asset": ["bed", "coffee_table", "door"], "object": ["vase", "ottoman", "lamp"]}, {"region": "Bathroom_2", "asset": ["bathroom_counter", "sink", "bathroom_cabinet"], "object": ["shower_cabin", "rug", "lamp"]}, {"region": "Kitchen_4", "asset": ["dishwasher", "oven", "stove"], "object": ["flower", "flower_vase", "bowl"]}, {"region": "Living_room_5", "asset": ["shelf", "couch", "tv"], "object": ["magazine", "box", "cushion"]}, {"region": "Hallway_6", "asset": ["door"], "object": ["lamp", "vent", "rack"]}, {"region": "Bathroom_7", "asset": ["toilet", "toilet_paper", "sink"], "object": ["towel", "soap_dish", "dustbin"]}, {"region": "Laundry_room_8", "asset": ["counter", "door", "bench"], "object":

```
        ["basket", "coat_hanger", "lamp"]}, {"region": "
        Lounge_Waiting_Room_9", "asset": ["chair", "table"], "object": ["
        flower_vase", "rack", "flatware"]}, {"region": "Entryway_Foyer_10
        ", "asset": ["door", "table"], "object": ["lamp"]}, {"region": "
        Bedroom_11", "asset": ["door", "bed", "dresser"], "object": ["
        plant", "lamp", "box"]}, {"region": "Bathroom_12", "asset": ["
        bathroom_counter", "toilet", "toilet_paper"], "object": ["
        shower_handle", "rug", "towel_bar"]}, {"region": "Bedroom_13", "
        asset": ["nightstand", "door", "bed"], "object": ["rug", "vent", "
        lamp"]}]}}
single agent task: {"phase_1": {"Single robot start position": {"
        robot_1": "Bedroom_1"}, "Single robot target object position": {"
        robot_1": "Bedroom_13"}, "Target object": "rug", "Single robot end
         position": {"robot_1": "Bathroom_2"}, "Single robot travel path":
         {"robot_1": ["Bedroom_1 -> Kichen_4", "Kichen_4 -> Living_room_5
        ", "Living_room_5 -> Hallway_6", "Hallway_6 -> Bathroom_12", "
        Bathroom_12 -> Bedroom_13", "Bedroom_13 -> Bathroom_12", "
        Bathroom_12 -> Hallway_6", "Hallway_6 -> Living_room_5", "
        Living_room_5 -> Kichen_4", "Kichen_4 -> Bedroom_1", "Bedroom_1 ->
         Bathroom_2"]},"Task instruction": "Take the rug in bedroom to the
         bathroom counter in Bathroom."}}
OUTPUT:

{
"phase_2": {
    "Collaborative robot start region": {
     "robot_1": "Bedroom_1",
     "robot_2": "Hallway_6"
    },
    "Collaborative type": "Type-A1",
    "Transfer region": "Kitchen_4",
    "Subtask instruction": {
     "robot_1": "Take the rug from shelf in Kitchen to the bathroom
         counter in Bathroom.",
     "robot_2": "Take the rug in bedroom to the shelf in Kitchen."
    },
    "Subtask list": {
     "robot_1": [
       "Move_to('shelf_4')",
       "Grab('rug')",
       "Move_to('bathroom_counter_2')",
       "Release('rug')"
     ],
     "robot_2": [
       "Move_to('rug_13')",
       "Grab('rug')",
       "Move_to('shelf_4')",
       "Release('rug')"
     ]
    }
  }
}
```

### A.11.2  SKILL FUNCTION

```
1  def check_collab_path_efficient_sim_graph(
2          robot_2_region,
3          transfer_nodes,
4          transfer_asset,
5          collab_type,
6          solo_cost,  #
7          r1_nodes,
```

```
 8              target_nodes,
 9              end_nodes,
10              region_objects,
11              G):
12          r2_nodes = cluster_center_node('_'+robot_2_region.split("_")[-1],
            ↪  nx.nodes(G), nx.get_node_attributes(G, 'position'))
13
14          if r2_nodes == None:
15              return {"efficient": False}
16
17          print("r2_nodes:", r2_nodes)
18
19          if transfer_nodes:
20              print("transfer_asset: ", transfer_asset)
21              transfer_asset_pos = find_target_position(region_objects,
                ↪  '_'+transfer_nodes.split("_")[-1], '
                ↪  '.join(transfer_asset.split('_')))
22              if transfer_asset_pos != None:
23                  transfer_nodes, _, _ =
                    ↪  nearest_navpoint_to_object_vec(transfer_asset_pos,
                    ↪  '_'+transfer_nodes.split("_")[-1], nx.nodes(G),
                    ↪  nx.get_node_attributes(G, 'position'))
24                  insert_temp_point("transfer_"+transfer_asset, G,
                    ↪  transfer_asset_pos, transfer_nodes)
25                  transfer_nodes = "transfer_"+transfer_asset
26
27                  if transfer_nodes != None:
28                      print("transfer_nodes", transfer_nodes)
29                  else:
30                      return {"efficient": False}
31              else:
32                  return {"efficient": False}
33          else:
34              return {"efficient": False}
35
36          if collab_type == "Type-A1":
37
38              try:
39                  r2_to_target = nx.shortest_path_length(G,
40                                          source=r2_nodes,
41                                          target=target_nodes,
42                                          weight='weight')
43              except (nx.NetworkXNoPath, nx.NodeNotFound):
44                  return {"efficient": False}
45
46              try:
47                  target_to_transfer = nx.shortest_path_length(G,
48                                          source=target_nodes,
49                                          target=transfer_nodes,
50                                          weight='weight')
51              except (nx.NetworkXNoPath, nx.NodeNotFound):
52                  return {"efficient": False}
53
54              robot2_first_leg = r2_to_target + target_to_transfer
55
56              try:
57                  r1_to_transfer = nx.shortest_path_length(G,
58                                          source=r1_nodes,
59                                          target=transfer_nodes,
60                                          weight='weight')
61              except (nx.NetworkXNoPath, nx.NodeNotFound):
62                  return {"efficient": False}
63
64              parallel_leg = max(robot2_first_leg, r1_to_transfer)
65
```

```
66              try:
67                  transfer_to_end = nx.shortest_path_length(G,
68                                          source=transfer_nodes,
69                                          target=end_nodes,

71              except (nx.NetworkXNoPath, nx.NodeNotFound):
72                  return {"efficient": False}

74              g_parallel_cost = parallel_leg + transfer_to_end
75              r1_parallel_cost = r1_to_transfer + transfer_to_end

77              path_infor = {
78                  "r2_to_target": r2_to_target,
79                  "target_to_transfer": target_to_transfer,
80                  "robot2_first_leg": robot2_first_leg,
81                  "r1_to_transfer": r1_to_transfer,
82                  "parallel_leg": parallel_leg,
83                  "transfer_to_end": transfer_to_end,
84                  "type": collab_type
85              }

87              print("g_rate: ", g_parallel_cost / solo_cost)
88              print("r1_rate: ", r1_parallel_cost / solo_cost)

90              return {
91                  "g_efficient": g_parallel_cost < solo_cost,
92                  "r1_efficient": r1_parallel_cost < solo_cost,
93                  "efficient": True if g_parallel_cost < solo_cost or
                     ↪  r1_parallel_cost < solo_cost else False,
94                  "g_rate": g_parallel_cost / solo_cost,
95                  "r1_rate": r1_parallel_cost / solo_cost,
96                  'path_info': path_infor
97              }

100         elif collab_type == "Type-A2":

102             try:
103                 r1_to_target = nx.shortest_path_length(G,
104                                         source=r1_nodes,
105                                         target=target_nodes,
106                                         weight='weight')
107             except (nx.NetworkXNoPath, nx.NodeNotFound):
108                 return {"efficient": False}

110             try:
111                 target_to_transfer = nx.shortest_path_length(G,
112                                         source=target_nodes,
113                                         target=transfer_nodes,
114                                         weight='weight')
115             except (nx.NetworkXNoPath, nx.NodeNotFound):
116                 return {"efficient": False}

118             robot1_first_leg = r1_to_target + target_to_transfer

120             try:
121                 r2_to_transfer = nx.shortest_path_length(G,
122                                         source=r2_nodes,
123                                         target=transfer_nodes,
124                                         weight='weight')
125             except (nx.NetworkXNoPath, nx.NodeNotFound):
126                 return {"efficient": False}

128             parallel_leg = max(robot1_first_leg, r2_to_transfer)
129
```

```
130            try:
131                transfer_to_end = nx.shortest_path_length(G,
132                                   source=transfer_nodes,
133                                   target=end_nodes,
134                                   weight='weight')
135            except (nx.NetworkXNoPath, nx.NodeNotFound):
136                return {"efficient": False}
137
138            g_parallel_cost = parallel_leg + transfer_to_end
139            r1_parallel_cost = r1_to_target + target_to_transfer
140
141            path_infor = {
142                "r1_to_target":r1_to_target,
143                "target_to_transfer": target_to_transfer,
144                "robot1_first_leg": robot1_first_leg,
145                "r2_to_transfer": r2_to_transfer,
146                "parallel_leg": parallel_leg,
147                "transfer_to_end": transfer_to_end,
148                "type": collab_type
149            }
150
151            print("g_rate: ", g_parallel_cost / solo_cost)
152            print("r1_rate: ", r1_parallel_cost / solo_cost)
153
154            return {
155                "g_efficient": g_parallel_cost < solo_cost,
156                "r1_efficient": r1_parallel_cost < solo_cost,
157                "efficient": True if g_parallel_cost < solo_cost or
                   ↪  r1_parallel_cost < solo_cost else False,
158                "g_rate": g_parallel_cost / solo_cost,
159                "r1_rate": r1_parallel_cost / solo_cost,
160                'path_info': path_infor
161            }
162
163    else:
164                return {
165                    "efficient": False
166                }
```

## A.12 PROMPTS FOR GRAPH CONTEXTUAL TYPING

---

**Room Type Reasoning**

**System:**
You are an AI assistant that specializes in spatial reasoning and semantic scene understanding. Your task is to infer the most likely type of a room (e.g., Bedroom, Bathroom, Kitchen, Laundry room, Living room, Hallway, Tie, Balcony, Terrace, etc.) based on the provided contextual clues.

**Rules:** There are two part of the input: neighbors and objects. The neighbors part includes information about the number and types of adjacent rooms. This gives spatial and functional context, which helps narrow down the likely use of the unknown room. The objects part lists the items found in the unknown room. These object types and frequencies are strong indicators of the room's function.

```
 1  When inferring the type of an unknown room, your reasoning must be
    ↪   guided by both the semantic distribution of objects and the
    ↪   spatial context of neighboring rooms.
 2  """
 3  - Always prioritize distinctive, functionally indicative objects over
    ↪   generic decorative ones.
 4  - Use neighboring room types to constrain plausible options (e.g., a
    ↪   room between two Bathrooms is unlikely to be a Kitchen).
 5  - Avoid inferring ambiguous multifunctional rooms unless object
    ↪   diversity strongly supports it.
 6  - Room type must be one from the predefined category list (e.g.,
    ↪   Bedroom, Bathroom, Kitchen, etc.).
 7  - In cases where the object list is sparse, weigh neighbor consistency
    ↪   more heavily.
 8  """
 9  Below is the full list of supported room types, each with a brief
    ↪   description of its primary function.
10  """
11  - Bedroom | used for sleeping and personal rest.
12  - Bathroom | supports hygiene tasks like bathing and toileting.
13  - Kitchen | designed for cooking and food preparation.
14  - Dining Room | a place for eating meals, often next to the kitchen.
15  - Living Room | used for leisure, social interaction, or
    ↪   entertainment.
16  - Study / Office | a workspace for reading, writing, or computer use.
17  - Laundry Room | contains appliances and tools for washing clothes.
18  - Closet / Storage Room | used to store clothes, tools, or household
    ↪   items.
19  - Hallway / Corridor | connects other rooms; mainly transitional.
20  - Garage | for parking vehicles or storing tools and equipment.
21  - Kids Room / Nursery | a bedroom tailored for children, often with
    ↪   toys.
22  - Balcony / Terrace | a semi-outdoor area for air, light, or drying.
23  - Media Room / Home Theater | equipped for movies or audio-visual
    ↪   activities.
24  - Gym / Fitness Room | contains equipment for exercise and physical
    ↪   training.
25  - Library | a quiet space for reading or storing books.
26  - Meeting Room | a formal area for group discussions or presentations.
27  - Lounge / Waiting Room | a rest area in public or semi-public
    ↪   buildings.
28  - Pantry / Bar | a compact space for storing or serving drinks/snacks.
29  - Dressing Room | dedicated to changing clothes or grooming.
30  - Entryway / Foyer | the front entrance space where people enter the
    ↪   home
31  """
32  Output format constraints:
33  """
34  - Only output the final predicted room type.
35  - The output must be a single Python dictionary with the format:
36    {"Unknown room type": "<Room Name>"}
```

---

```
37  - Do NOT include any reasoning, explanation, or analysis in the
    ↪    output.
38  - Do NOT output multiple room types or probabilities|only the most
    ↪    likely one.
39  - Do NOT include any commentary, bullet points, or markdown
    ↪    formatting.
40  """
41  Here is an example of the INPUT and OUTPUT:
42  INPUT:
43  ```
44  neighbors: "There are 2 neighboring rooms, belonging to 1 type,
    ↪    including 2 Kitchens."
45  objects: "The unknown room contains 77 objects, belonging to 30 types.
    ↪    The top 5 most frequent items are: 22 photos, 11 chairs, 3 plants,
    ↪    3 vases, and 3 shelfs."
46  ```
47  OUTPUT:
48  ```
49  {"Unknown room type": "Living room"}
50  ```
```

## A.13  ABLATION FOR PROMPTS

### A.13.1  DIRECTLY TWO-AGENT TASK GENERATION

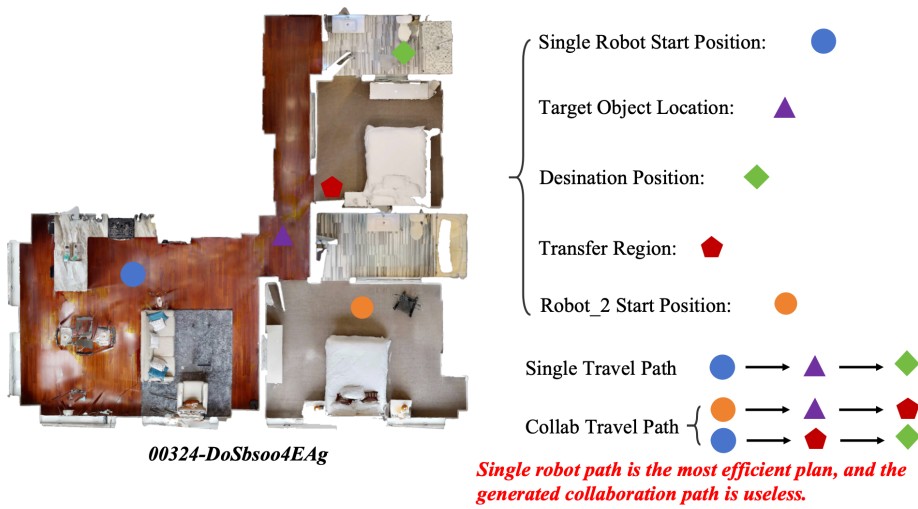

Figure 17: Common failure case of directly generating a two-agent task.

In fact, the initial versions of CoNavBench were exactly based on directly generating two-agent tasks, as you suggested, and our current "single-agent task then split" pipeline is the result of several iterations driven by empirical observations. Concretely, we implemented and tested three major versions of the data-generation pipeline on a fixed test set of HM3D environments (00087-YY8rqV6L6rf, 00299-bdp1XNEdvmW, 00323-yHLr6bvWsVm, 00324-DoSbsoo4EAg, 00444-sX9xad6ULKc, 00612-GsQBY83r3hb) to allow controlled comparisons under identical prompts. However, after extensive testing, we found that both V1 and V2 suffered from very low-quality multi-robot collaboration, mainly due to the difficulty of long-horizon spatial reasoning with two agents at once. Typical failure modes included: I) Little or no true parallelism: the LLM often produced plans where one robot did almost all the work while the other remained idle, so executing the plan with a single robot sequentially was actually more efficient, as shown in Figure 17. II) Inconsistent or invalid coordination: mismatches between the transfer region and the subtask descriptions,

missing handover steps, or paths that did not align with the scene layout. Hence, motivated by these observations, we adopt the two-stage NavCraft design. We have added this discussion in the Supplements Section for more details.

---

**Directly Two-agent Task Rule Prompt Version 1**

```
1   Important rules regarding the scene graph:
2   """
3   - You must leverage the scene graph topology to design meaningful
    ↪   multi-robot cooperation.
4   - The connectivity between regions (provided in the "link" field of
    ↪   the scene graph) must be used to ensure:
5       - Robots do not start in adjacent regions.
6       - Task handoffs (e.g., object relay between robots) happen in
        ↪   regions that are connected.
7   - The selection of regions and object transfer paths should reflect
    ↪   realistic spatial planning based on the graph.
8   - Encourage designs where the first robot delivers an object to an
    ↪   intermediate node (transfer zone), and the second robot continues
    ↪   from there. This creates natural cooperation patterns.
9   """
10  There are something you need to pay attention to:
11  """
12  - A robot should not start in the same region where it needs to pick
    ↪   up or drop off objects.
13  - Each robot's contribution should reduce the overall task cost:
14      - Avoid assigning tasks that could be completed more efficiently
        ↪   by a single robot.
15      - Only use multi-robot handoff when it significantly reduces
        ↪   travel distance or enables parallelism.
16  - The multi-robot plan you generate should not be less efficient than
    ↪   a single-robot plan for the same task. Cooperation should lead to
    ↪   either reduced execution time or more balanced workload across
    ↪   agents.
17  - The objects involved must be portable and must appear in the input
    ↪   scene.
18  - The task must involve only 1 to 2 different regions.
19  - The region IDs (e.g., "Kitchen_1", "Bedroom_3") do not imply spatial
    ↪   proximity or connectivity. Only the "link" field in the scene
    ↪   graph provides valid region-to-region connections. Do not assume
    ↪   regions with similar names or IDs are connected.
20  - The full task must contain 4 to 6 subtasks total (across all
    ↪   robots).
21  - The task you generate should be similar to instructions like "Take
    ↪   an object in one region to a certain place in one region."
22  - Subtasks must follow logical ordering: a robot must Move_to before
    ↪   Grab or Release; it cannot Release without having grabbed the
    ↪   object first.
23  - The region mentioned in Move_to() should match the region mentioned
    ↪   in the high-level instruction.
24  - The task must reflect multi-robot cooperation, such as transporting
    ↪   an object to a place where another robot picks it up and
    ↪   continues.
25  - Do not use low-level action terms like "grab", "release", or
    ↪   "move_to" in "Task instruction" or "Subtask instruction".
26  """
27  Your output should be a Python dictionary with the following keys:
28  """
29  - "Robots position": A dictionary assigning each robot to a starting
    ↪   region. Each robot must start in a different region, and the
    ↪   regions must be at least one hop apart in the scene graph.
30  - "Task instruction": A conversational high-level instruction
    ↪   describing the overall collaborative task.
```

```
31  - "Transfer position":  A list of intermediate regions (e.g., region
    ↪   IDs) that serve as **handover zones** between robots. These
    ↪   positions should be selected from the scene graph and must be
    ↪   reachable by both the sender and the receiver robot. Use them to
    ↪   support collaborative efficiency: for example, Robot A may carry
    ↪   an object from Region X to a transfer position, where Robot B
    ↪   picks it up and delivers it to the final destination. When
    ↪   selecting a transfer position, prefer regions that minimize the
    ↪   total travel distance between participating robots.
32  - "Subtask instruction": A dictionary giving each robot a
    ↪   natural-language description of its individual role in the task.
    ↪   Avoid technical terms like "grab" or "release".
33  - "Subtask list": A dictionary mapping each robot to a list of
    ↪   low-level subtasks. These subtasks should be composed of the
    ↪   following actions:
34    - Move_to("object_region_id"): Walk to an object or location in a
      ↪   region. The "object_region_id" must combine the object name and
      ↪   its region ID from the input scene (e.g., "lamp_1").
35    - Grab("object"): Pick up the object. The robot must first move to
      ↪   the object's location. The object must exist in the "objects"
      ↪   list of the input scene.
36    - Release("object"): Place the object in the target asset or
      ↪   location. The robot must first move to the target. The object
      ↪   must exist in the input scene.
37  """
38  Make sure the task instruction conversational enough, and the task
    ↪   should reasonable.
```

### Directly Two-agent Task Rule Prompt Version 2

```
1   Important rules regarding scene and task planning:
2   """
3   - Use the region connectivity graph ("link") to determine all movement
    ↪   and transfer feasibility.
4   - Robots must start in **different**, **non-adjacent** regions.
5   - A robot cannot start in the same region where it will pick up or
    ↪   drop off an object.
6   - Task handoff must happen in a region reachable from both the sender
    ↪   and receiver.
7   - A robot must Move_to a region before performing Grab or Release.
8   - A robot must Grab an object before it can Release it.
9   - The region specified in Move_to() must match the one implied in the
    ↪   instruction (no teleportation).
10  - Subtasks must follow this logical order: Move_to → Grab → Move_to →
    ↪   Release.
11  - The objects involved must:
12    - Be listed in the input scene's "object" field (i.e., exist and be
      ↪   portable).
13    - Be manipulated **only**, not assets.
14  - Subtasks must not violate logical flow or act on unavailable
    ↪   objects/assets.
15  - The overall plan must involve **1 to 2 regions total** (e.g., task
    ↪   origin, destination, or transfer area).
16  - Tasks must be split into **4 to 6 total low-level subtasks**, across
    ↪   all robots.
17  - Task planning should reflect genuine cooperation:
18    - Prefer parallelism or reduced path cost through collaboration.
19    - Avoid plans where a single robot could complete the task more
      ↪   efficiently.
20    - Multi-robot plans **must not be less efficient** than single-robot
      ↪   alternatives.
```

```
21  - The task must exhibit **explicit collaboration**, such as robot_1
    ↪  transporting an object to an intermediate location, and robot_2
    ↪  completing delivery.
22  - Avoid assuming proximity based on region name:
23    - Region IDs (e.g., "Kitchen_1", "Kitchen_2") **do not imply spatial
    ↪  adjacency**.
24    - Use only "link" data to determine connectivity.
25  - Task instruction and subtask instructions should:
26    - Be natural-language (e.g., "Take the object from A to B").
27    - **Avoid technical terms** like "grab", "release", or "move_to".
28  """
29  Region Symbol Definitions (used in output and reasoning):
30  """
31  - Region_A: Initial position of robot_1
32  - Region_B: Location of the portable object
33  - Region_X: Handoff region (object relay from robot_1 to robot_2)
34  - Region_Y: Initial position of robot_2
35  - Region_C: Final destination for object delivery
36  """
37  Spatial Constraints:
38  """
39  - Region_A and Region_Y must NOT be adjacent.
40  - Region X and Region B must NOT be adjacent.
41  - Region X and Region C must NOT be adjacent.
42  - Among Region_A, Region_X, Region_Y, and Region_C, Region_B must be
    ↪  the closest to Region_A based on the region connectivity graph
    ↪  (i.e., shortest path length from Region_A).
43  - Among Region_A, Region_X, Region_Y, and Region_B, Region_C must be
    ↪  the closest to Region_Y based on the region connectivity graph
    ↪  (i.e., shortest path length from Region_Y).
44  """
45  Output Format:
46  """
47  {
48    "Robots start position": {"robot_1": "Region_A", "robot_2":
    ↪  "Region_Y"},
49    "Transfer position": "Region_X",
50    "Robot travel path": {
51      "robot_1": ["Region_A -> Region_B", "Region_B -> Region_X"],
52      "robot_2": ["Region_Y -> Region_X", "Region_X -> Region_C"]
53    },
54    "Task instruction": "Take the [object_B] from Region_B to the
    ↪  [asset_C] in Region_C.",
55    "Subtask instruction": {
56      "robot_1": "Take [object_B] in Region_B to the [asset_X] in
    ↪  Region_X.",
57      "robot_2": "Take [object_B] from [asset_X] in Region_X to the
    ↪  [asset_C] in Region_C."
58    },
59    "Subtask list": {
60      "robot_1": ["Move_to('object_B')", "Grab('object')",
    ↪  "Move_to('asset_X')", "Release('object')"],
61      "robot_2": ["Move_to('asset_X')", "Grab('object')",
    ↪  "Move_to('asset_C')", "Release('object')"]
62    }
63  }
64  """
65  Output field explanation:
66  """
67  - "Robots start position": Dict mapping each robot to its starting
    ↪  region (must follow spatial constraints).
68  - "Transfer position": Region ID (or list of region IDs) where object
    ↪  is passed from robot_1 to robot_2.
```

```
69  - "Robot travel path": Dict showing robot movement as ordered
    ↪  region-to-region transitions.
70  - "Task instruction": Natural description of full multi-robot delivery
    ↪  task.
71  - "Subtask instruction": A dictionary giving each robot a
    ↪  natural-language description of its individual role in the task.
    ↪  Avoid technical terms like "grab" or "release".
72  - "Subtask list": Dict of robot action sequences using these
    ↪  primitives:
73      - Move_to("object_region_id")
74      - Grab("object")
75      - Release("object")
76  """
77  Make sure the task instruction conversational enough, and the task
    ↪  should reasonable.
```

