# OpenReview forum: "CoNavBench: Collaborative Long-Horizon Vision-Language Navigation Benchmark"
_ICLR.cc/2026/Conference — ICLR 2026 Poster_

### Official Review · Reviewer_wsr9 · 2025-10-27

**Soundness:** 3
**Presentation:** 3
**Contribution:** 3
**Rating:** 6
**Confidence:** 4

**Summary:**

This paper introduces CoNavBench, a large-scale benchmark designed to evaluate collaborative vision-and-language navigation (VLN) under multi-agent, long-horizon settings. The authors further propose NavCraft, a graph-grounded data generation and validation platform that constructs semantically annotated scene graphs, generates single-agent base tasks (NavCraft-S), and lifts them into collaborative multi-robot schedules (NavCraft-C). The experiments use Qwen2.5-VL-3B/7B models fine-tuned on the dataset, demonstrating improved performance over single-robot baselines and validating the benefit of collaboration in task success and path efficiency. Overall, this work establishes an interesting benchmark for multi-agent embodied navigation by unifying data generation, simulation, and evaluation.

**Strengths:**

The paper convincingly identifies a major gap in the VLN landscape: the lack of standardized evaluation for multi-agent, cooperative scenarios, and fills it with a well-structured benchmark and taxonomy.
NavCraft’s design (scene-graph generation, task validation, and efficiency tools) is technically detailed, modular, and reproducible within Habitat-Sim, enabling scalable and context-aware task generation.
The framework successfully merges semantic graph annotation, spatial reasoning, and LLM-driven language prompts to produce feasible, diverse, and verifiable tasks.
Finetuned Qwen2.5-VL models show consistent improvements in SR, SPL, and CSR across both single- and multi-agent settings, confirming the utility of CoNavBench for training collaborative policies.

**Weaknesses:**

*Limited collaboration scope*

Only two fixed handoff types (A1, A2) are explored, which may not fully capture the diversity of real collaborative behaviors (e.g., concurrent exploration, dynamic task reassignment).

*Data generation*

The dependence on GPT-4o-mini and closed APIs limits long-term reproducibility; no quantitative analysis is provided on the variability or bias of generated instructions.

*Lack of detailed time-efficiency evaluation*

Although collaboration is claimed to reduce makespan, this paper needs more clarification on collaboration acceleration and time-based metrics.

*Incomplete ablations for important modules*

Components such as the memory-aware mechanism, efficiency tool library, and profile-conditioned sampling are presented but not individually quantified in their contributions.

**Questions:**

Besides the weakness mentioned above, the reviewer has some extra questions below:

How does the benchmark handle multi-agent interference and collision during simulation? Are these events explicitly annotated or only indirectly measured by task failure?

Could the authors provide a quantitative comparison between NavCraft-generated and manually designed tasks to justify realism and linguistic fidelity?

Why does the Qwen2.5-VL-3B outperform 7B in several collaborative tasks? Does this stem from optimization saturation or data scarcity?

---

> ### Author Response · Authors · 2025-11-20
> **Response to Reviewer wsr9 - Stage 1**
>
> **W1: Only two fixed handoff types (A1, A2) are explored, which may not fully capture the diversity of real collaborative behaviors (e.g., concurrent exploration, dynamic task reassignment).**
>
> A1: We really appreciate for this comment and apologize for the confusion. In our formulation, the two handoff types A1 and A2 are intended as **minimal atomic collaboration patterns** for object-relay tasks: who starts from the user side, who approaches the object side, and where the handoff occurs. Richer behaviors (e.g., concurrent exploration or dynamic task reassignment) can be realized by **re-planning** on top of these atoms: given the updated agent states and a new user instruction, NavCraft-C can be invoked again to produce a new relay plan (handoff region + subtask allocation), effectively supporting dynamic collaboration rather than being restricted to a single, static A1/A2 choice.
>
> Regarding the number of agents, most existing VLN benchmarks focus on a single robot, and CoNavBench is, **to our knowledge, the first benchmark explicitly targeting collaborative navigation**. We therefore start with the simplest non-trivial setting of two robots, which already requires explicit coordination and handoff reasoning. In addition, the indoor HM3D environments have limited spatial extent; in our experiments, two robots are sufficient to saturate most tasks, while scaling to three or more robots would demand substantially more complex instructions and larger or more structured scenes. Extending beyond the current A1/A2 patterns and beyond two agents is a natural direction for future work built on top of our framework. We also addressed this limitation in the manuscript on Page 9.
>
> **W2: The dependence on GPT-4o-mini and closed APIs limits long-term reproducibility; no quantitative analysis is provided on the variability or bias of generated instructions.**
>
> A2: We really appreciate for raising this important point. For the dependence on GPT-4o-mini and closed APIs, **please kindly refer to our Public Comments**, where we describe (i) releasing the full data-generation pipeline and prompts, and (ii) additional experiments with an open-source model (Qwen2.5-32B-Instruct) showing that CoNavBench is not tied to a single proprietary backend. Regarding the variability and bias of generated instructions, our design explicitly constrains the output space to minimize uncontrolled LLM variation. Concretely, NavCraft uses a simple, **fixed grammar for task instructions of the form** “*Take the [object] from [target object region] to the [asset] in [end region].*” which is strictly grounded in the scene graph (objects, regions, and assets) and described in Appendix A.10.1 on Page 23. We keep the slot-filling structure fixed, so the linguistic variability is limited to the specific object/region choices dictated by the environment rather than open-ended phrasing. This greatly reduces both stylistic variability and potential bias in how instructions are expressed, while preserving the semantic diversity induced by different scenes, objects, and goal regions.
>
> If users of the benchmark desire more linguistic variety (e.g., “*Could you please take …*”, “*Please move …*”), this can be added in a **post-processing** step by paraphrasing the canonical template, without changing the underlying grounded tuples.
>
> **W3: Although collaboration is claimed to reduce makespan, this paper needs more clarification on collaboration acceleration and time-based metrics.**
>
> A3: We really appreciate your construtive suggestions. we added time-based analyses to better clarify “collaboration acceleration.” On the CoNavBench test set, we compare the single-robot execution of each mission with the two-robot collaborative plans generated by NavCraft under the same high-level instruction. We find that, on average, the main robot’s effective working time is **reduced by 20.63%** in the collaborative setting. This means collaboration not only shortens the overall makespan, but also frees the main robot to spend more time interacting with the user or handling other complex tasks.

---

> ### Author Response · Authors · 2025-11-20
> **Response to Reviewer wsr9 - Stage 2**
>
> **W4: Components such as the memory-aware mechanism, efficiency tool library, and profile-conditioned sampling are presented but not individually quantified in their contributions.**
>
> A4: We apologize for the confusion. The memory-aware mechanism is not our contribution, but comes directly from the LH-VLN backbone that we adopt. **We already cite LH-VLN in the manuscript**, and in the revised manuscript, we **have removed** “memory-aware mechanism” from the abstract and conclusion to avoid overstating it as a new component. For this reason, we do not run a separate ablation on this module.
>
> For the efficiency tool library, we now provide ablations by selectively enabling/disabling the tools in NavCraft-S and NavCraft-C. Here, “1” means the tool library is enabled for that stage, and “0” means it is disabled:
>
> | Tool Library (NavCraft-S / NavCraft-C) | Success Rate (Single / Collab) | Collab Gain Avg |
> |:--:|:----:|:---:|
> | 0 / 1                                  | 43% / 41.86%                    | 25.75%           |
> | 1 / 0                                  | 65% / 6.15%                     | 21.53%           |
> | 0 / 0                                  | 38% / 18.42%                    | 26.62%           |
>
> These ablations show that removing the efficiency tools from either NavCraft-S or NavCraft-C degrades either single-agent success, collaborative success, or collaboration gain, indicating that both stages benefit from the efficiency-aware design.
>
> For the profile-conditioned sampling, we provide the analysis of its contribution (instruction and object diversity, role-specific demand patterns); please kindly refer to our response to Reviewer 7PLf, A6 and the corresponding section in Appendix A.8 on Page 19.
>
> **Q1: How does the benchmark handle multi-agent interference and collision during simulation? Are these events explicitly annotated or only indirectly measured by task failure?**
>
> A5: Thanks for your insightful questions. Our benchmark is defined at the **high-level planning** layer (task decomposition, role assignment, relay region selection), and does not introduce a new low-level collision-avoidance module. In simulation, collision and obstacle avoidance are handled by the underlying navigation stack (e.g., Habitat-Sim controllers, existing VLN models, or any traditional navigation / multi-agent planner the user chooses to plug in). CoNavBench is therefore agnostic to the specific low-level controller: we evaluate whether the high-level collaborative plan is reasonable, while concrete collision handling is delegated to the chosen navigation algorithm.
>
> **Q2: Could the authors provide a quantitative comparison between NavCraft-generated and manually designed tasks to justify realism and linguistic fidelity?**
>
> A6: Thanks for your insightful suggestion. To quantitatively compare NavCraft-generated tasks with human-designed ones in terms of realism and collaboration structure, **we conducted a small case study, more detailed in Appendix A.7 on Page 18.** Specifically, we invite three volunteers. For three randomly selected HM3D scenes (*IDs 00712, 00539, 000732*), we provided each participant with the same scene graph and high-level single-robot mission as used by NavCraft, along with a bird’s-eye-view rendering of the environment.
>
> For each scene, participants were asked to 1) *choose an ideal transfer region* and 2) *select a collaboration type (Type-A1 vs. A2) that they considered most reasonable for a two-robot execution*. We then compared their choices to NavCraft-C’s automatically generated transfer region and collaboration type.
>
> We observe that 1) **Scene 00712**: NavCraft and all three participants chose the **same transfer region and the same Type-A1 pattern**; 2) **Scene 00539**: NavCraft’s transfer region and type matched the choices of two participants (User A and User C). The remaining participant (User B) deliberately selected a relay region closer to the main robot, preferring a design where the helper robot completes as much of the physical delivery as possible so that the **main robot can spend more time near the user for interaction.** Interestingly, this “*user-interaction–prioritized*” preference also appears in User B’s choice for scene 000732, suggesting a consistent personal strategy rather than disagreement with the task semantics. 3) **Scene 000732**: Both NavCraft and the human annotators favored parallelism-first strategies, but they **diverged on the exact relay region**: human participants, with access to the BEV image, tended to select the wider kitchen bar as a safer, more spacious handoff location, while NavCraft, operating only on the *symbolic scene graph without asset footprint information*, selected a narrower bar table near the window as the transfer region. This mismatch is therefore attributable to geometric detail (asset size / usable surface) that is **not represented in the current graph abstraction**, rather than a failure of logical reasoning about the task.

---

> ### Author Response · Authors · 2025-11-20
> **Response to Reviewer wsr9 - Stage 3**
>
> **Q3: Why does the Qwen2.5-VL-3B outperform 7B in several collaborative tasks? Does this stem from optimization saturation or data scarcity?**
>
> A7: We really appreciate your insightful question. Our current understanding is that this phenomenon is mainly due to **data scarcity** and under-training of the larger model, rather than optimization saturation.
>
> In our experiments, both Qwen2.5-VL-3B and 7B are fine-tuned on exactly the same CoNavBench trajectories. The absolute number of training samples is relatively modest for full-parameter VLN fine-tuning at the 7B scale; based on our preliminary scaling experiments, we estimate that a 7B VLN model would typically require on the order of 5k–10k diverse trajectories to consistently outperform a 3B model across collaborative tasks. With the current dataset size, the 3B model appears to be more sample-efficient, while the 7B model does not fully realize its capacity and can even overfit to spurious patterns in certain settings. This aligns with cases where 3B slightly outperforms 7B in collaborative scenarios.
>
> We therefore attribute the observed gap primarily to limited data rather than an inherent weakness of larger models. As future work, we plan to expand CoNavBench by **generating more trajectories in additional high-quality 3D environments**. One promising direction we are exploring is to leverage open **3D Gaussian Splatting (3D-GS) indoor datasets** such as InteriorGS [1], convert them to mesh (.glb) format, and import them into the Habitat-Sim simulator. We also note that the visual fidelity of the current HM3D-based Habitat renderings is lower than that of recent 3D-GS datasets, so migrating to or augmenting with 3D-GS–based environments may benefit both perception and planning.
>
> [1] InteriorGS: A 3D Gaussian Splatting Dataset of Semantically Labeled Indoor Scenes. ArXiV 2025.

---

### Official Review · Reviewer_UuGT · 2025-10-29

**Soundness:** 3
**Presentation:** 2
**Contribution:** 2
**Rating:** 4
**Confidence:** 3

**Summary:**

1. CoNavBench Benchmark: It includes 4048 single-robot and multi-robot collaborative tasks, designed to test how robot collaboration can optimize task completion time and improve efficiency in long-horizon tasks.

2. NavCraft Platform: An automated data generation platform for collaborative tasks that uses the scene graph from Habitat-Sim to create semantically rich environmental layouts and generate collaborative tasks. It includes two stages of task generation: NavCraft-S (for generating single-robot tasks) and NavCraft-C (for generating collaborative tasks).

3. Collaborative Models and Performance Evaluation: Experiments using the Qwen2.5-VL-3B model show that the collaborative model improves step-level success by 18.11% and reduces task completion time compared to the single-robot model.

**Strengths:**

By introducing multi-robot collaboration, the CoNavBench benchmark optimizes the completion time of long-horizon tasks. Compared to single-robot tasks, collaborative robots can perform multiple subtasks simultaneously, effectively reducing task delays and idle time, significantly improving overall task efficiency. The CoNavBench benchmark includes 4048 single-robot and multi-robot collaborative tasks, providing a comprehensive evaluation of collaborative navigation systems in complex tasks. It tests how robots coordinate and distribute tasks in scenarios where multiple robots work together.

**Weaknesses:**

1. The inference speed of large models is very slow; will there be a significant delay in task completion?
2. Will the two agents collide in the environment?
3. There are many VLN methods based on large models [1]; why were they not compared with these methods?
4. Are the initial positions of the two agents the same, or is one agent directly summoned?




[1] Cheng, An-Chieh, et al. "Navila: Legged robot vision-language-action model for navigation." arXiv preprint arXiv:2412.04453 (2024).

**Questions:**

same as weakness.

---

> ### Author Response · Authors · 2025-11-20
> **Response to Reviewer UuGT- Stage 1**
>
> **W1: The inference speed of large models is very slow; will there be a significant delay in task completion?**
>
> A1: We really appreciate you for this question and apologize for the confusion. CoNavBench is designed as a **high-level planning and benchmarking framework**, and in our current setup the NavCraft is only invoked once per episode to generate the collaborative plan before navigation starts. The subsequent execution is carried out by some amazing downstream VLN methods or other navigation controllers.
>
> **W2: Will the two agents collide in the environment?**
>
> A2: Thanks for your insightful discussion. Our benchmark is defined at the high-level planning layer (task decomposition, role assignment, relay region selection), and **does not introduce a new low-level collision-avoidance module**. In simulation, collision and obstacle avoidance are handled by the underlying navigation stack (e.g., Habitat-Sim controllers, existing VLN models, or any traditional navigation / multi-agent planner the user chooses to plug in). **CoNavBench is therefore agnostic to the specific low-level controller**: we evaluate whether the high-level collaborative plan is reasonable, while concrete collision handling is delegated to the chosen navigation algorithm.
>
> **W3: There are many VLN methods based on large models [1]; why were they not compared with these methods?**
>
> A3: We apology for missing such amazing large model based VLN methods and **have added it in our Related Work**. From our perspective, note by **Reviewer cERd** “*Do the authors believe this difference would disappear with SOTA large language models?*”. We agree that part of the performance gap between single-robot and two-robot settings can be attributed to the fact that each robot in the collaborative case receives a **shorter, more localized subtask, which is easier for current VLN to handle**. Hypothetically, if future state-of-the-art VLN models could robustly solve long-horizon single-agent missions end-to-end from user instructions, we would indeed expect the success-rate advantage of decomposing the task across multiple robots to shrink. However, even with such powerful models, we believe there are **inherent advantages of collaboration that do not disappear**, most notably in terms of time efficiency and user experience. Multi-robot collaboration enables true spatial and temporal parallelism (e.g., robots moving and preparing different parts of the environment simultaneously), which a single agent, no matter how capable, cannot replicate. In CoNavBench, we have added experiments show that collaborative planning yields an **average time-efficiency gain of 20.63%** compared to the single-robot execution, highlighting that the benefit is not only about feasibility but also about reduced robot task time.
> And we are also trying our best to utilize the Navila[1] into our CoNavBench setting. **Please kindly refer to A3.2.**
>
> **W4: Are the initial positions of the two agents the same, or is one agent directly summoned?**
>
> A4: We really appreciate this insightful question. In our benchmark, the two agents **do not share the same initial position**, nor is one “summoned” from the main robot’s location. Instead, their start positions are sampled as distinct regions in the environment according to our planning JSON / prompt rules (see Appendix A.11.1 on Page 29), so the robots are spatially distributed from the beginning of each episode. NavCraft then takes these initial positions as input and decides whether collaboration is beneficial; if so, it plans a relay by selecting a transfer region and a handoff type (A1 or A2) consistent with the two start locations.

---

> ### Author Response · Authors · 2025-11-27
> **Response to Reviewer UuGT- Stage 2**
>
> **W3: There are many VLN methods based on large models [1]; why were they not compared with these methods?**
>
> A3.2:  We followed the official NaVILA installation guide to build Habitat v0.1.7 from source and downloaded the released pretrained checkpoint. Using the R2R inference scripts provided by NaVILA, we verified our reproduction: the metrics are closely aligned with those reported in the NaVILA paper, indicating that our reproduction is reliable.
>
> | NE    | OS    | SR    | SPL   |
> |:-----:|:-----:|:-----:|:-----:|
> | 5.21  | 61.55 | 53.83 | 49.25 |
>
> Since NaVILA **only provides inference interfaces for R2R on MP3D**, while our proposed CoNavBench is built on HM3D, we implemented a conversion script that transforms CoNavBench episodes into the R2R data format expected by NaVILA. An example converted episode is shown below:
>
> ```json
> [
>   {
>     "episode_id": 1,
>     "trajectory_id": 1,
>     "scene_id": "hm3d/SgkmkWjjmDJ/SgkmkWjjmDJ.basis.glb",
>     "start_position": [6.428913116455078, -2.556666374206543, 0.45417889952659607],
>     "start_rotation": [0.0, -0.7071067811865475, 0.0, 0.7071067811865476],
>     "instruction": {
>       "instruction_text": "Move forward through the living room to reach the hanger.",
>       "instruction_tokens": [1411, 915, 2228, 2202, 1290, 1842, 2246, 1775, 2202, 1054, 9, 0, 0, ...]
>     },
>     "goals": [
>       {
>         "position": [0.8125040531158447, -2.556666374206543, -0.04697234928607941],
>         "radius": 3.0
>       }
>     ],
>     "reference_path": [
>       [6.178913116455078, -2.556666374206543, 0.4541788697242737],
>       ...
>     ],
>     "info": {
>       "geodesic_distance": 5.500386360516943
>     }
>   }
> ]
> ```
> During inference, we encountered a Habitat simulator–induced scene loading issue, which has also been reported in the official NaVILA repository (GitHub issue #25 https://github.com/AnjieCheng/NaVILA/issues/25). At the time of writing, there is no confirmed fix from the community. Nevertheless, since the **core contribution** of CoNavBench is to *decompose long-horizon navigation tasks into multiple short-horizon sub-tasks over multiple robots to improve the SR and thereby improve parallel execution efficiency*, we try our best to further analyze NaVILA’s R2R inference results to study the relationship between task distance and success rate. The results show that as the task distance increases, the success rate (SR) consistently decreases, and the success rate for long-distance tasks is effectively 0.
>
> | Task Distance (m)      | [3, 6) | [6, 9) | [9, 12) | [12, 15) | [15, 18) | [21, +∞) |
> |:----------------------:|:------:|:------:|:-------:|:--------:|:--------:|:---------:|
> | SR                     | 60.98  | 56.56  | 49.90   | 47.31    | 41.67    | 0.00      |
>
> **This observation highlights the intrinsic difficulty of long-horizon navigation and motivates our design of multi-robot collaborative execution in CoNavBench, which helps improve the overall SR, particularly for long-distance tasks.**

---

### Official Review · Reviewer_7PLf · 2025-11-01

**Soundness:** 3
**Presentation:** 2
**Contribution:** 3
**Rating:** 4
**Confidence:** 2

**Summary:**

The paper introduces CoNavBench, a collaborative long-horizon Vision-and-Language Navigation (VLN) benchmark designed to study multi-robot cooperation under language-guided tasks. It extends prior single-agent datasets by supporting multi-agent task decomposition, handoff scheduling, and rendezvous-based cooperation. To generate the data, the authors build NavCraft, a graph-grounded generation platform with semantic scene graphs, hierarchical planning agents (NavCraft-S / NavCraft-C), and an on-graph efficiency tool library for validation and optimization. Experiments demonstrate reduced makespan and improved step-level success over single-agent baselines using Qwen2.5-VL policies.

**Strengths:**

1. CoNavBench provides a large-scale benchmark (4048 episodes) covering both single and multi-robot navigation with rich annotations for collaboration types and performance metrics.
2. The proposed NavCraft system establishes a structured data-generation pipeline grounded in semantic graphs, incorporating task synthesis, collaboration lifting, and efficiency validation, which enhances data consistency and scalability.
3. The framework introduces a useful utility for checking reachability, congestion, and timing constraints directly on scene graphs, which is a practical contribution for large-scale synthetic task design.
4. The manuscript provides dataset statistics, training configurations, and baselines clearly, allowing others to replicate and evaluate under consistent protocols.

**Weaknesses:**

1. Visual clarity is not acceptable at all, for example, Figure 9 is difficult to interpret and provides almost no new information beyond stating that the training loss decreases. It lacks analytical insight and does not meaningfully support the main claims. The visual examples fail to illustrate collaborative interaction or path optimization clearly. Comparisons are blurry or incomplete. An example is the visualization of collaborative cases, the authors didn't provide any useful information in these cases to support the claim.
2. While the data-generation pipeline heavily relies on prompt engineering for NavCraft, there is no systematic or theoretical analysis of its design choices, prompt sensitivity, or ablation, leaving uncertainty about robustness.
3. The entire framework and experiments are conducted in simulation (e.g., Habitat-3). Although acceptable at this stage, no real-world experiments or transfer evaluations are included to verify deployability or generalization.

**Questions:**

1. Could authors provide more clear visualization results to clarify the claims, for example, how Figure 9 contributes analytically, whether it reflects convergence stability, mode collapse, or loss behavior differences between different models?
2. Are there plans to improve qualitative visualizations to more clearly demonstrate cooperative efficiency or multi-agent coordination?
3. Have you analyzed how different prompt templates or role-conditioning strategies affect the generated tasks’ diversity or validity?
4. Is there any plan to evaluate NavCraft or CoNavBench tasks on real robots or real-world scans to assess practical applicability?

**Details Of Ethics Concerns:**

There is no ethics concern from the reviewer.

---

> ### Author Response · Authors · 2025-11-20
> **Response to Reviewer 7PLf - Stage 1**
>
> **W1: Visual clarity is not acceptable at all, for example, Figure 9 is difficult to interpret and provides almost no new information beyond stating that the training loss decreases. It lacks analytical insight and does not meaningfully support the main claims. The visual examples fail to illustrate collaborative interaction or path optimization clearly. Comparisons are blurry or incomplete. An example is the visualization of collaborative cases, the authors didn't provide any useful information in these cases to support the claim.**
>
> A1: We sincerely apologize for the confusion caused by Figure 9 and thank you for pointing this out. Our intention with this figure was **not to claim any deep analytical insight from the loss curves themselves**, but simply to demonstrate that CoNavBench-generated data can be used to successfully fine-tune Qwen-series LLM models. In Figure 9, we use the same training objective as LH-VLN [1] : **the blue curve corresponds to the long-horizon loss, while the red curve corresponds to the step-by-step loss.**
> We agree that training-loss plots alone do not support our core contributions. Our central claims are: (i) CoNavBench provides a planning-centric benchmark where a high-level planner decomposes a single-agent long-horizon mission into collaborative multi-robot subtasks, and (ii) such collaboration improves both success rate and time efficiency compared to single-robot execution. As discussed elsewhere in the rebuttal, we show that multi-robot collaboration yields an average 20.63% parallel time-efficiency gain over the single-robot baseline, and real-robot experiments in an exhibition-hall scenario further indicate improved user experience (Please kindly refer to Public Comment A2).
>
> We appreciate the your feedback on the visual clarity of our collaborative examples. After carefully reviewing the manuscript, we found that the original version of Figure 5 indeed did not clearly convey how collaboration and path optimization work. In the revised version, **we have explicitly marked the transfer region and their planned paths on the ego view.**
>
> **W2: While the data-generation pipeline heavily relies on prompt engineering for NavCraft, there is no systematic or theoretical analysis of its design choices, prompt sensitivity, or ablation, leaving uncertainty about robustness.**
>
> A2: We really appreciate this suggestion. As also raised by Reviewer cERd, **we have added a more systematic discussion and empirical analysis of the NavCraft prompt design**. In Appendix A.13 on Page 38, we now describe and compare two major earlier prompt versions (V1 and V2) that directly generate two-agent plans, and contrast them with our final two-stage design. Because the full prompts are long, **we additionally release all development versions of the NavCraft-S and NavCraft-C prompts in the supplementary codes**, so that their evolution is **fully transparent**. Each version was evaluated on a fixed set of HM3D scenes (please refer to our response to Reviewer cERd, A5), and iteratively refined to improve CoT robustness in terms of stable collaborative task generation (*valid transfer region, consistent subtask allocation, and parallel execution*).
>
> **W3: The entire framework and experiments are conducted in simulation (e.g., Habitat-3). Although acceptable at this stage, no real-world experiments or transfer evaluations are included to verify deployability or generalization.**
>
> A3: Please kindly refer to Public Comment A2.
>
> **Q1: Could authors provide more clear visualization results to clarify the claims, for example, how Figure 9 contributes analytically, whether it reflects convergence stability, mode collapse, or loss behavior differences between different models?**
>
> A4: Please kindly refer to, and we have fixed the misunderstanding of Figure 9 and Figure 5, and added the real robot demo video in the Anonymous Website: https://navcraft.github.io.
>
> **Q2: Are there plans to improve qualitative visualizations to more clearly demonstrate cooperative efficiency or multi-agent coordination?**
>
> A5: We really thanks for your constructive suggestions. We have added the **real robot collaboration visualization on the Appendix A.9 Page 21.**

---

> ### Author Response · Authors · 2025-11-20
> **Response to Reviewer 7PLf - Stage 2**
>
> **Q3: Have you analyzed how different prompt templates or role-conditioning strategies affect the generated tasks’ diversity or validity?**
>
> A6: We really appreciate your constructive suggestions. **We added experiments to quantitatively evaluate the impact of user profiles on task realism and diversity**. We utilize five distinct roles: Role 1: 25-year-old single male PhD student; Role 2: 33-year-old married female lawyer; Role 3: 65-year-old married retired male; Role 4: 9-year-old single male student; Role 5: 20-year-old single female undergraduate.
>
> For comparison, we also include a no-profile baseline. On a set of 20 scenes, we fix the random seed and scene configuration, and only vary whether a user profile is provided (and, if so, which role). NavCraft therefore receives identical environment inputs, and the difference comes solely from the presence and type of user profile. I) Overall diversity. As shown in Figure 14 on Appendix A.8 Page 20, adding user profiles increases both instruction and object diversity. Across the 20 scenes, the total number of unique instructions grows from **316 (no profile) to 367 (with profiles)**, a **16.14%** increase. The total number of unique pickup objects increases from **98** to **115**, a **17.35%** increase. On a per-scene basis, the average number of unique instructions rises from **18.22 to 21.11**, corresponding to a **15.86% increase**. II) Role-specific task demand. With profiles enabled, different roles exhibit **clearly different object-demand patterns** in the same scene, reflecting their underlying persona. As illustrated in Figure 15 on Appendix A.8 Page 20, for example in scene 00495-CQWES1bawee, Role 3 (retired) tends to request practical items such as *hand soap for cleaning*, Role 4 (child) prefers *playful or visually attractive objects* such as pictures, and Role 5 (young female undergraduate) is more likely to request *decorative items* like flowers.
>
> These results indicate that conditioning on user profiles not only **increases overall instruction and object diversity**, but also induces meaningful, **role-dependent variations in task demand**, which we believe makes CoNavBench more realistic for **user-centric multi-robot navigation**.
>
> **Q4: Is there any plan to evaluate NavCraft or CoNavBench tasks on real robots or real-world scans to assess practical applicability?**
>
> A7: We really appreciate your constructive suggestions. Please kindly refer to Public Comment A2.

---

### Official Review · Reviewer_cERd · 2025-11-01

**Soundness:** 3
**Presentation:** 3
**Contribution:** 3
**Rating:** 6
**Confidence:** 3

**Summary:**

The authors introduce CoNavBench, a new benchmark for "Collaborative Long-Horizon Vision-and-Language Navigation" where multiple robot agents collaborate to follow instructions inside 3D simulated homes (using Habitat-Sim). They introduce Navcraft, an algorithm to generate these multi-robot cooperation scenes where agents share and hand off subtasks. It contains 4048 episodes annotated with scene graphs, but more can be generated by Navcraft. Navcraft uses a two-stage hierarchy, Navcraft-S and NavCraft-C, to first generate a single-robot task, then break it down into a two-robot collaborative task, with graph-based scene understanding and validation. The authors fine-tune Qwen2.5-VL models on this benchmark, reporting improvements over single-agent baselines.

**Strengths:**

This is a novel benchmark for robotic cooperation in 3D home environments. The proposed mehtod can generate an arbitrary number of scenarios. The paper presents a clear and interesting pipeline (NavCraft-S and NavCraft-C) for task creation, graph annotation and validation. It is overall clear with illustrative figures. The authors use standard metrics for measuring performance on the benchmark, and provide a thorough baseline and ablation analysis with various LLM APIs.

**Weaknesses:**

- The benchmark, while innovative, is still limited to two agents and specific relay-style tasks (one robot carries something from A to B, hands it off to a second robot to carry it from B to C). Dynamic collaborations are not explored.
- NavCraft relies on proprietary LLM APis for data generation, which can lead to issues for reproducibility of the benchmarking. This limitation is addressed by the authors.
- Some parts are dense in formulas and could benefit from a high-level intuitive explanation beforehand.

**Questions:**

- Can the author expand on how do the results vary based on the chosen collaboration type (A1 vs A2) ?
- How does this approach (generate single-agent task then split) compare against directly generating a two-agent task?
- Authors report using two robots for the task instead of a single one improves performance. This can be explained by each robot being able to learn its task more efficiently. Do the authors believe this difference would disappear with SOTA large language models?

---

> ### Author Response · Authors · 2025-11-20
> **Response to Reviewer cERd - Stage 1**
>
> **W1: The benchmark, while innovative, is still limited to two agents and specific relay-style tasks (one robot carries something from A to B, hands it off to a second robot to carry it from B to C). Dynamic collaborations are not explored.**
>
> A1: We really appreciate your insightful comment. I) CoNavBench is specifically designed to study **how an initial natural-language instruction from the user can be converted into a complete high-level plan for multiple robots**, including subtask decomposition, role assignment, and the selection of transfer regions, such that the overall task can be completed without further human intervention. For this reason, the benchmark currently instantiates a two-robot relay setting as the simplest non-trivial form of collaboration that still requires explicit coordination and handover reasoning. II) While dynamic collaborations driven by new user instructions during execution are not the primary focus of the benchmark, **our framework can in fact support such scenarios via re-planning**. After partial execution, one can provide the updated start positions of robot 1 and robot 2 to NavCraft-C, which then generates a new subtask list and a new transfer region. In this way, NavCraft-C can re-decide both the handoff location and the type of relay, thereby enabling dynamic collaboration. **Please kindly refer to Appendix A.11 on Page 29 and provide the corresponding implementation in the released source code in the supplementary material.** III) Regarding the limitation to two agents, we note that most existing VLN settings focus on a single robot, and CoNavBench is, **to the best of our knowledge, the first benchmark targeting collaborative multi-robot navigation.** We therefore intentionally adopt a relatively simple two-robot configuration as a first step. In addition, the indoor environments provided by HM3D have limited spatial extent, and our experiments suggest that two robots are sufficient to saturate most indoor tasks; scaling beyond two agents would require substantially more complex language instructions and larger or more structured environments. Extending CoNavBench to more than two robots and to richer, non-relay-style collaborative patterns is a natural and important direction for future work built on top of our formulation.
>
> **W2: NavCraft relies on proprietary LLM APis for data generation, which can lead to issues for reproducibility of the benchmarking. This limitation is addressed by the authors.**
>
> A2: We really appreciate your constructive question. Please kindly refer to the Public Comments.
>
> **W3: Some parts are dense in formulas and could benefit from a high-level intuitive explanation beforehand.**
>
> A3: Sorry for the confusion. We have revised the Section3.1.1’s Graph Contextual Typing, Section3.1.2’s Feasibility and Section3.1.3’s Collaborative Generation. i) Section3.1.1’s Graph Contextual Typing added part goes like this “*After the preceding steps, a small fraction of regions may still be typed as Unknown. Intuitively, these are ambiguous areas where local object votes and connectivity cues are not confident enough on their own.*” on Page 5 Line 238. ii) Section3.1.2’s Feasibility added part goes like this “*Given a candidate triple (s,t,e), NavCraft-S must first ensure that the underlying navigation problem is meaningful: each leg should be reachable on the region graph and long enough to span multiple rooms.*” on Page 5 Line 259. iii) Section3.1.3’s Collaborative Generation added part goes like this “*We quantify when it is worthwhile to involve a second agent by comparing how much travel load the main agent would bear alone versus under different collaboration patterns. Intuitively, a collaboration is only accepted if bringing in the helper strictly shortens the main agent's own route.*” on Page 6 Line 287.

---

> ### Author Response · Authors · 2025-11-20
> **Response to Reviewer cERd - Stage 2**
>
> **Q1: Can the author expand on how do the results vary based on the chosen collaboration type (A1 vs A2) ?**
>
> A4: We really appreciate your suggestions. We have conducted the step-by-step Tasks performance comparison on different collaborative Type A1 and A2. The performance gap bewteen A1 and A2 is on bar. Overall, the two collaboration types exhibit very similar trends across all metrics, with A2 showing slightly lower SR/SPL/ISR and a slightly higher NE. Importantly, these differences are moderate and do not change any of our main conclusions: the relative ranking of methods and the benefits of collaboration remain stable under both A1 and A2. We have added this per-type analysis and table to the revised manuscript supplement to make the effect of the collaboration type more explicit.
>
> | Method        | Collaborative Type | SR    | SPL   | ISR    | NE    |
> |:-------------:|:----------------:|:-----:|:-----:|:------:|:-----:|
> | Qwen-2.5VL-3B | A1               | 34.47 | 17.41 | 34.47  | 5.78  |
> | Qwen-2.5VL-3B | A2               | 29.09 | 15.58 | 29.09  | 6.91  |
>
> **Q2: How does this approach (generate single-agent task then split) compare against directly generating a two-agent task?**
>
> A5: We really appreciate your insightful questions. In fact, the initial versions of CoNavBench were exactly based on directly generating two-agent tasks, as you suggested, and **our current “single-agent task then split” pipeline is the result of several iterations driven by empirical observations.** Concretely, we implemented and tested three major versions of the data-generation pipeline on a fixed test set of HM3D environments *(00087-YY8rqV6L6rf,00299-bdp1XNEdvmW,00323-yHLr6bvWsVm,00324-DoSbsoo4EAg,00444-sX9xad6ULKc,00612-GsQBY83r3hb)* to allow controlled comparisons under identical prompts.
>
> **Version 1** (direct two-agent generation, details on Appendix A.13 on Page 38): The LLM directly outputs initial positions for both robots, a transfer region, a global task instruction, and per-robot subtask/action sequences, e.g., Robots start position, Transfer position, Task instruction, Subtask instruction, Subtask list. The example of output json goes like:
> ```json
> {"Robots start position": {"robot_1": "Hallway_6","robot_2": "Bedroom_1"},
>  "Transfer position": "Kitchen_4",
>  "Task instruction": "take the lamp in bedroom to the bathroom counter in Bathroom",
>  "Subtask instruction": {"robot_1": "take the lamp in bedroom to the shelf in Kitchen", "robot_2": "take the lamp in Kitchen to the bathroom counter in Bathroom" },
>  "Subtask list": [
>     {"robot_1": ["Move_to('lamp_13')","Grab('lamp')","Move_to('shelf_4')","Release('lamp')"]},
>     {"robot_2": ["Move_to('lamp_4')","Grab('lamp')", "Move_to('bathroom_counter_2')","Release('lamp')"]}]}
> ```
> **Version 2** (direct two-agent generation + travel paths, details on Appendix A.13 on Page 38): Building on V1, we further asked the LLM to generate explicit travel paths for each robot (a “Robot travel path”field describing region-to-region moves), while still jointly producing start positions, transfer region, subtasks, and action lists. The example of output json goes like:
>
> ```json
> {"Robots start position": {"robot_1": "Hallway_6","robot_2": "Bedroom_1"},
>  "Transfer position": "Kitchen_4",
>   "Robot travel path": {
>     "robot_1": ["Hallway_6 -> Bedroom_13", "Bedroom_13 -> Kitchen_4"],
>     "robot_2": ["Bedroom_1 -> Kitchen_4", "Kitchen_4 -> Bathroom_2"]
>   },
>   "Task instruction": "take the lamp in bedroom to the bathroom counter in Bathroom",
>   "Subtask instruction": {
>     "robot_1": "take the lamp in bedroom to the shelf in Kitchen",
>     "robot_2": "take the lamp from shelf in Kitchen to the bathroom counter in Bathroom" },
>  "Subtask list": [
>     {"robot_1": ["Move_to('lamp_13')","Grab('lamp')","Move_to('shelf_4')","Release('lamp')"]},
>     {"robot_2": ["Move_to('lamp_4')","Grab('lamp')", "Move_to('bathroom_counter_2')","Release('lamp')"]}]}
> ```
> However, after extensive testing, we found that both V1 and V2 suffered from **very low-quality multi-robot collaboration**, mainly due to the difficulty of long-horizon spatial reasoning with two agents at once. Typical failure modes included: I) **Little or no true parallelism**: the LLM often produced plans where one robot did almost all the work while the other remained idle, so executing the plan with a single robot sequentially was actually more efficient, as shown in Figure 17 in Appendix A.13 on Page 38. II) Inconsistent or invalid coordination: mismatches between the transfer region and the subtask descriptions, missing handover steps, or paths that did not align with the scene layout. Hence, motivated by these observations, we adopt **the two-stage NavCraft design**. We have added this discussion in the Supplements Section for more details.

---

> ### Author Response · Authors · 2025-11-20
> **Response to Reviewer cERd - Stage 3**
>
> **Q3: Authors report using two robots for the task instead of a single one improves performance. This can be explained by each robot being able to learn its task more efficiently. Do the authors believe this difference would disappear with SOTA large language models?**
>
> A6: Thanks for your thoughtful discussion. We agree that part of the performance gap between single-robot and two-robot settings can be attributed to the fact that each robot in the collaborative case receives a shorter, more localized subtask, which is easier for current large-language-model based VLN to handle. Hypothetically, **if future state-of-the-art VLN models could robustly solve long-horizon single-agent missions end-to-end from user instructions**, we would indeed expect the success-rate advantage of decomposing the task across multiple robots to shrink.
>
> However, even with such powerful models, we believe there are **inherent advantages of collaboration that do not disappear**, most notably in terms of **time efficiency and user experience**. Multi-robot collaboration enables true spatial and temporal parallelism (e.g., robots moving and preparing different parts of the environment simultaneously), which a single agent, no matter how capable, cannot replicate. In CoNavBench, we have added experiments show that collaborative planning **yields an average time-efficiency gain of 20.63% compared to the single-robot execution**, highlighting that the benefit is not only about feasibility but also about reduced robot task time.
>
> Furthermore, during rebuttal stage, we also added real-world experiment in an exhibition-hall scenario, we observe that multi-robot execution improves the interactive experience, instead of a single robot executing a long plan while the **user passively waits**, multiple robots can move concurrently, provide intermediate feedback, and keep the environment “visibly active”, which users perceive as more engaging and responsive (see also Public Comment A2). Thus, while better VLN model may narrow the success-rate gap, we expect the time-efficiency and interaction advantages of multi-robot collaboration to remain significant.

---

### Author Response · Authors · 2025-11-20
**Public Comments**

We thank all reviewers for their constructive comments on our work. We found following comments that were common amongst more than one reviewer, hence we highlight them here.

**Q1: “NavCraft relies on proprietary LLM APis for data generation, which can lead to issues for reproducibility of the benchmarking. This limitation is addressed by the authors.” [Reviewer cERd]; “The dependence on GPT-4o-mini and closed APIs limits long-term reproducibility” [wsr9].**

A1: We fully agree that relying solely on proprietary LLM APIs could hinder strict reproducibility. To address this concern, we have taken the following steps: Open-sourcing code and prompts. **All our code is open-sourced**, and the complete implementation (including prompts) is provided in the supplementary material. This allows the community to audit our pipeline and to substitute any backbone model they prefer, ensuring that NavCraft can be continuously updated as new models appear.
Powered with open-source LLM. To further improve reproducibility, we additionally evaluate NavCraft using the open-source Qwen2.5-32B-Instruct model [1], hosted by a third-party provider. We strictly reuse the same prompts and random seeds as in Table 4 of the paper. The results are summarized below:

| Powered Agent          | Success Rate (Single / Collab) | Collab Gain (Max / Avg) | Cost ($) | Sample Eff. (s/iter) |
|:----------------------:|:------------------------------:|:------------------------:|:--------:|:---------------------:|
| Qwen2.5-32B-Instruct   | 35% / 8.57%                    | 14.58% / 10.28%          | 2.505    | 43.57                 |


We have added these results into the latest version of the manuscript. As can be seen, while open-sourced Qwen2.5-32B-Instruct achieves competitive performance, the GPT-4o-mini family from OpenAI still provides the best trade-off between effectiveness and cost in our setting. This is consistent with other recent embodied-navigation benchmarks such as NavRAG [2] and LH-VLN [3], which also rely on the GPT-4o series as their backbone LLM. These modifications make our benchmark substantially more reproducible while preserving its practical relevance for the community.

[1] https://huggingface.co/Qwen/Qwen2.5-32B-Instruct

[2] NavRAG: Generating User Demand Instructions for Embodied Navigation through Retrieval-Augmented LLM. ACL 2025 Findings.

[3] Towards long-horizon vision-language navigation: Platform, benchmark and method. CVPR 2025.

**Q2: Is there any plan to evaluate NavCraft or CoNavBench tasks on real robots or real-world scans to assess practical applicability?**

A2: We really appreciate your suggestion. We conducted the real robot collaboration case study in the exhibition hall. The process can be summarized as following: I) **3D Reconstruction of the Real Exhibition Hall and BEV View**, we first capture multi-view images of the real exhibition hall and perform 3D scene reconstruction using a COLMAP with 3DGS pipeline. This yields a high-fidelity 3D model of the environment and a top-down BEV (bird’s-eye view), which facilitates intuitive visualization and inspection of the exhibition space. II) **Demand-Oriented Navigation for a Single-Robot Water-Fetching Task**. During the exhibition tour, the user reports feeling thirsty and requests water, triggering a demand-oriented navigation task. After receiving the ‘fetch water’ command, the single robot must navigate to the target object, a bottle of water, located far away on the opposite side of the hall and perform the fetching operation. III) **User Waiting Problem Caused by Long Round-Trip Paths**. After picking up the water, the robot has to traverse the long path back to the user’s location. Because the overall round trip is time-consuming, the user can only passively wait during this period, leading to poor user experience and boredom. IV) **NavCraft-C High-Level Task Planner and Multi-Robot Relay Design**. To address the low efficiency and unsatisfying user experience in long-horizon single-robot tasks, we propose the NavCraft-C high-level task planner. It introduces a second robot and defines a transfer region in the exhibition hall, enabling task relay and parallel execution, thereby improving overall task efficiency and user experience. V) **Multi-Robot Task Relay and Improved User Interaction**. While robot 2 is heading to fetch the water, robot 1 stays with the user, continuously providing explanations and interaction to maintain engagement. When robot 2 approaches the transfer region, robot 1 proactively moves to this region to receive the bottle and then hands the water to the user, achieving efficient multi-robot collaboration and an enhanced human-robot interaction experience. **And we have update the corresponding real robot demo video section in the Anonymous Website: https://navcraft.github.io for better understanding.**

---

### Author Response · Authors · 2025-11-30
**Summary Comments to Area Chair**

Dear Area Chair,

We appreciate the time and effort invested by the reviewers in providing constructive feedback on our submission. In the following, we first summarize the reviewer’s recognition of our work and highlight the main contribution of the paper. Then, we summarize the key points raised by the reviewers, along with our responses and the corresponding revisions incorporated into the manuscript.

---

We are encouraged that all Reviewers (cERd, 7PLf, UuGT, wsr9) recognize this work as a novel benchmark filling a gap in multi-agent VLN evaluation.
- NavCraft pipeline is praised by cERd, 7PLf, and wsr9 for its clear, structured, and scalable semantic-graph–based task generation.
- Efficiency validation tools (reachability, congestion, timing) are highlighted by 7PLf and wsr9 for their practical value.
- Clarity and reproducibility of dataset statistics, baselines, and experimental setup are commended by cERd, 7PLf, and wsr9.
---

We highlight our contributions as follows:
- To the best of our knowledge, the first benchmark for collaborative multi-robot navigation.
- A scalable task-generation pipeline (NavCraft) supporting synthesis, collaboration, and automatic validation.
- Reproducible baselines and open-source protocols demonstrating gains from collaboration.

---

We summarize the key suggestions and concerns raised by the reviewers, followed by our detailed and the corresponding revisions into the manuscripts.

- **Real robot experiments**

As suggested by Reviewer 7PLf and UuGT, 1) We have update the corresponding real robot demo video section in the Anonymous Website: https://navcraft.github.io for better understanding, the parallel collaboration helps improved the user interaction; 2) We also conduct experiment on analyze NaVILA inference results to study the relationship between task distance and success rate. The results show that as the task distance increases, the success rate (SR) consistently decreases, and the success rate for long-distance tasks is effectively 0.

| Task Distance (m) | [3, 6) | [6, 9) | [9, 12) | [12, 15) | [15, 18) | [21, +∞) |
|:--:|:--:|:----:|:----:|:---:|:--:|:---:|
| SR | 60.98 | 56.56 | 49.90 | 47.31 | 41.67| 0.00  |

This highlights the challenge of long-horizon navigation and motivates CoNavBench’s multi-robot collaboration, which boosts SR, especially for long-distance tasks.

- **Open-source LLM for NavCraft**

As suggested by Reviewer cERd and wsr9, we have taken the following steps: 1) Open-sourcing code and prompts. All our code is open-sourced, and the complete implementation (including prompts) is provided in the supplementary material. 2) We additionally evaluate NavCraft using the open-source Qwen2.5-32B-Instruct model. We strictly reuse the same prompts and random seeds as in Table 4 of the manuscripts.

| Powered Agent | Success Rate (Single / Collab) | Collab Gain (Max / Avg) | Cost ($) | Sample Eff. (s/iter) |
|:---:|:-------:|:---:|:-----:|:-----:|
| Qwen2.5-32B-Instruct | 35% / 8.57% | 14.58% / 10.28% | 2.505| 43.57  |

- **Ablation on task generation mechanism**

As suggested by Reviewer cERd, we added a discussion in Appendix A.13 (Page 38) comparing single-agent-then-split versus direct two-agent generation. Experiments show that the two-stage design produces higher-quality multi-robot collaboration by avoiding low parallelism and coordination inconsistencies seen in direct two-agent generation.

- **Ablation on efficiency tool library**

As suggested by Reviewer wsr9, we added ablations by selectively enabling or disabling the efficiency tools in NavCraft-S and NavCraft-C. The results show that removing these tools from either stage reduces single-agent success, collaborative success, or collaboration gain, demonstrating that both stages benefit from the efficiency-aware design.

| Tool Library (NavCraft-S / NavCraft-C) | Success Rate (Single / Collab) | Collab Gain Avg |
|:--:|:----:|:--:|
| 0 / 1 | 43% / 41.86% | 25.75%  |
| 1 / 0 | 65% / 6.15%  | 21.53%  |
| 0 / 0 | 38% / 18.42% | 26.62%  |

- **Ablation on user-profile condition**

As suggested by Reviewers 7PLf and wsr9, we evaluated the impact of user profiles on task realism and diversity (Appendix A.8, Page 20, Figure 14). Adding user profiles increases both instruction diversity (316 → 367, +16.14%) and object diversity (98 → 115, +17.35%), while producing distinct object-demand patterns across roles, reflecting underlying persona differences.

- **Case study on manually vs. NavCraft on designing collaboration task**

As suggested by Reviewer wsr9, a case study (Appendix A.7, Page 18) shows that NavCraft-generated collaboration tasks largely align with human-designed ones, with minor differences due to personal preferences or geometric details.

- **Representations and visual Figures**

As suggested by Reviewers cERd and 7PLf, we improved the text in Sections 3.1.1–3.1.3 and revised Figures 5 and 9 to enhance readability and clarity.

---

### Meta-Review · Area_Chair_Lu2N · 2026-01-03

**Summary:**

I will list the most important comments that the reviewers noted during the review process:
1) The benchmark is still limited to two agents and specific relay-style tasks. Only two fixed handoff types (A1, A2) are explored, which may not fully capture the diversity of real collaborative behaviors.
2) NavCraft relies on proprietary LLM APis for data generation, which can lead to issues for reproducibility of the benchmarking.
3) There is no systematic or theoretical analysis of its design choices, prompt sensitivity, or ablation.
4) No real-world experiments or transfer evaluations are included to verify deployability or generalization.
5) The inference speed of large models is very slow.
6) There is no comparision with many VLN methods based on large models.
7) Components such as the memory-aware mechanism, efficiency tool library, and profile-conditioned sampling are presented but not individually quantified in their contributions.

**Reviewer Concerns:**

The authors did a lot of work during the rebuttal phase and addressed most of the reviewers' comments:
1) Proprietary LLM: authors additionally evaluate NavCraft using the open-source Qwen2.5-32B-Instruct model.
2) Ablations: authors conducted a large number of additional ablation experiments.
3) Real-world experiments: authors updated the corresponding real robot demo video section.

The authors agreed with some of the shortcomings or they remained unaddressed:
1) The benchmark is still limited to two agents and specific relay-style tasks.
2) The inference speed of large models is very slow.
3) There is no comparision with many VLN methods.
However, these remarks can be considered less significant.

**Reviewer Scores:**

1) cERd (score 6) would most likely have left his initial score.
2) 7PLf (score 4) could raise his score.
3) UuGT (score 4) would most likely have left his initial score.
4) wsr9 (score 6) would most likely have left his initial score.

---

### Decision · Program_Chairs · 2026-01-26

Accept (Poster)